# Molecular engineering of piezoelectricity in collagen-mimicking peptide assemblies

Santu Bera [1], Sarah Guerin [2], Hui Yuan[3], Joseph O'Donnell [2], Nicholas P. Reynolds[4,5], Oguzhan Maraba[2], Wei Ji [1], Linda J. W. Shimon [6], Pierre-Andre Cazade[2], Syed A. M. Tofail[2], Damien Thompson [2✉], Rusen Yang [3✉] & Ehud Gazit [1✉]

Realization of a self-assembled, nontoxic and eco-friendly piezoelectric device with high-performance, sensitivity and reliability is highly desirable to complement conventional inorganic and polymer based materials. Hierarchically organized natural materials such as collagen have long been posited to exhibit electromechanical properties that could potentially be amplified via molecular engineering to produce technologically relevant piezoelectricity. Here, by using a simple, minimalistic, building block of collagen, we fabricate a peptide-based piezoelectric generator utilising a radically different helical arrangement of Phe-Phe-derived peptide, Pro-Phe-Phe and Hyp-Phe-Phe, based only on proteinogenic amino acids. The simple addition of a hydroxyl group increases the expected piezoelectric response by an order of magnitude ($d_{35} = 27$ pm V$^{-1}$). The value is highest predicted to date in short natural peptides. We demonstrate tripeptide-based power generator that produces stable max current >50 nA and potential >1.2 V. Our results provide a promising device demonstration of computationally-guided molecular engineering of piezoelectricity in peptide nanotechnology.

[1] George S. Wise Faculty of Life Sciences, Shmunis School of Biomedicine and Cancer Research, Tel Aviv University, Ramat Aviv, Israel. [2] Department of Physics, Bernal Institute, University of Limerick, Limerick, Ireland. [3] School of Advanced Materials and Nanotechnology, Xidian University, Xi'an, China. [4] ARC Training Centre in Biodevices, Faculty of Science, Engineering and Technology, Swinburne University of Technology, Melbourne, Victoria, Australia. [5] Department of Chemistry and Physics, La Trobe Institute for Molecular Science, La Trobe University, Melbourne, Victoria, Australia. [6] Department of Chemical Research Support, Weizmann Institute of Science, Rehovot, Israel. ✉email: Damien.Thompson@ul.ie; rsyang@xidian.edu.cn; ehudg@post.tau.ac.il

Piezoelectric materials generate electrical energy in response to mechanical deformation. Piezoelectric devices are commonly made from a variety of inorganic materials and organic polymers[1–5], which limits their deployment in health monitoring and regenerative medicine due to reliance on toxic starting metals, complicated synthesis procedures, weak oxidation stability and poor sustainability[6]. Green piezoelectric materials that satisfy the requirements of ultra-high mechanosensitivity, flexibility and durability would provide a promising route to biocompatible multifunctional smart energy harvesters. Piezoelectricity has been widely observed in several natural materials including bone, collagen, viruses, cellulose and chitosan[7–10]. However, the piezoelectric response exhibited by these biomaterials is typically in the range of 0.1–10 pm V$^{-1}$, which is low for many potential applications[11]. Among these various piezoelectric biopolymers, collagen exhibits useful chemical and physical properties of extensibility, high tensile strength and swelling[12,13]. Moreover, the piezoelectricity of collagen may play a pivotal role in controlling bone growth[14], with fibrillar rat tail collagen exhibiting the highest measured shear $d_{14}$ value for a biomaterial of 12 pm V$^{-1}$ (pC N$^{-1}$)[15].

Biomimetics provides potentially disruptive advances in design and fabrication of novel functional materials. Using a minimalistic approach, self-assembled short peptides have emerged as versatile building blocks due to their inherent biocompatibility and highly engineerable properties that provide tailored functionality[16–19]. Although nanostructures formed by ultra-short peptide sequences predominantly exhibit β-sheet organization[20–22], the collagen structure is helical[23]. The large number of directionally aligned hydrogen bonds in the helical structure creates a macroscopic dipole that can couple with external electric fields and shear force to produce the piezoelectric response of collagen[24]. Examination of all collagen residues has demonstrated the post-translationally modified amino acid hydroxyproline (Hyp) to show the highest piezoelectric response[25]. Yet, by itself, Hyp does not exhibit the fibrillar assembly and helical structural pattern of collagen, and it has only very recently proved possible to design ultra-short peptides that can mimic the collagen supramolecular architecture[26]. On the other side, earlier findings suggested that the dynamic interactions of aromatic amino acid side chains in peptide-based structures can increase electrical conductivity[27,28]. The aromatic Phe-Phe-based β-sheet-rich biomaterials have been investigated and utilized for fabrication of green energy harvester[29,30]. However, tailor-made design of ultra-short peptide to achieve high piezoelectric response similar to collagen through mimicking its supramolecular architecture has remained elusive.

Molecular modelling, in particular density functional theory (DFT), has emerged as an effective tool for predicting and rationalizing the piezoelectric response of materials, from classic inorganic crystals[31,32] to polymers[33] and novel two-dimensional structures[34,35]. Our previous work has utilized DFT to accurately predict the elastic and piezoelectric tensors of amino acid[10,11], peptide[25] and biomineral crystals[36]. DFT can be used to complement techniques such as piezoresponse force microscopy (PFM)[37,38] or used in standalone predictive modelling studies of nanoscale electromechanical phenomena[39]. Classical molecular dynamics (MD) simulations are also widely used to study the kinetics of piezoelectric systems[40] and are particularly effective at studying systems in a liquid environment[41] and over a range of temperatures[42].

Here, aiming to engineer a minimalistic collagen-mimicking short peptides, we develop radically different organization of Phe-Phe-derived short peptide based only on natural amino acid by including both Hyp and aromatic Phe moieties in the sequence. Both Pro-Phe-Phe and Hyp-Phe-Phe tripeptides assemble into a helical-like sheet that is stabilized by the dry hydrophobic zipper interface of Phe residues. The self-assembled fibrillar biomaterial composed of helical-like molecular arrangement of the tripeptides exhibits excellent mechanical strength and high piezoelectric response (see benchmarks in Supplementary Table 1), which is maximized in the strong H-bonding Hyp variant. We designed the material to exhibit markedly higher piezoelectricity, which is achieved by modulating the electromechanical response of the tripeptide via side-chain engineering to optimize supramolecular polarization. When used as the active component in a power generator, the helical tripeptides show much larger short-circuit current and open-circuit voltage output as compared to β-sheet-rich peptide. Our findings demonstrate the rational modulation of peptide self-assembly to create tailored functionality and mark a significant step forward in molecular engineering of peptide piezoelectricity for nanotechnology applications by exemplifying the importance of targeting both primary and secondary structure.

## Results and discussion

**Characterization of Hyp-Phe-Phe assemblies**. To engineer collagen-mimicking natural piezoelectric short peptides, we chose the self-assembling tripeptide, Pro-Phe-Phe (Fig. 1a), an ultra-short helical natural peptide with high mechanical stability similar to that of collagen[26,43]. We substituted Pro for Hyp (Fig. 1a), as it exerts the highest piezoresponse among the collagen component amino acids[25]. Building on the recently solved solution state morphology and secondary structure of Pro-Phe-Phe[26], we used Fourier transform infrared (FTIR) and circular dichroism (CD) spectroscopy techniques to characterize the solution secondary structure of Hyp-Phe-Phe. The FTIR spectra showed a sharp amide I peak at 1645 cm$^{-1}$ with a shoulder at 1680 cm$^{-1}$, indicating the predominant helical structure (Fig. 1b) similar to Pro-Phe-Phe. The small peak shift towards lower wavenumber (compared with the standard helical structure, which usually falls near 1650–1655 cm$^{-1}$) agrees with the predicted frequency shift for ultra-short helices[44]. The CD spectrum strongly supported the FTIR data, as it exhibited double negative maxima, characteristic of helical conformations (Fig. 1c). The maxima at 210 and 230 nm were faintly red shifted compared to canonical double helix, as expected for shorter length peptides[45].

The supramolecular assembly of Hyp-Phe-Phe was explored using atomic force microscopy (AFM) and transmission electron microscopy (TEM) (Fig. 1d, e and Supplementary Figs. 1 and 2). The images revealed that the tripeptide self-assembled into uniform high aspect ratio fibres of 500 nm in diameter that extended for several micrometres (aspect ratio, L/D > 500). Single-crystal X-ray diffraction structures revealed the favourable molecular level interactions that direct the supramolecular organization (Fig. 1f). The asymmetric unit of the Hyp-Phe-Phe crystal comprised two peptide molecules sharing common structural features[26]. The torsion angles of the Phe$_2$ moiety were found to be localized within the right-handed helical region of the Ramachandran plot, with φ2 and ψ2 values of −71.1°, −70.5° and −43.2°, −41.9°, respectively, comparable to that of the Pro-Phe-Phe crystal. In the crystallographic $b$-direction, the adjacent molecules were connected through head-to-tail intermolecular H-bonds, generating an extended helical-like molecular organization (Fig. 1f, left and right illustrations). Neighbouring helical-like structural modules connected in a parallel orientation with the interface of the dimer stabilized by the aromatic zipper structure built from π–π interactions between the Phe side chains (Fig. 1f, middle illustration). Such a dry steric zipper interface has long been assumed to provide mechanical rigidity to amyloid fibres[46] and may provide additional stability to the Hyp-Phe-Phe

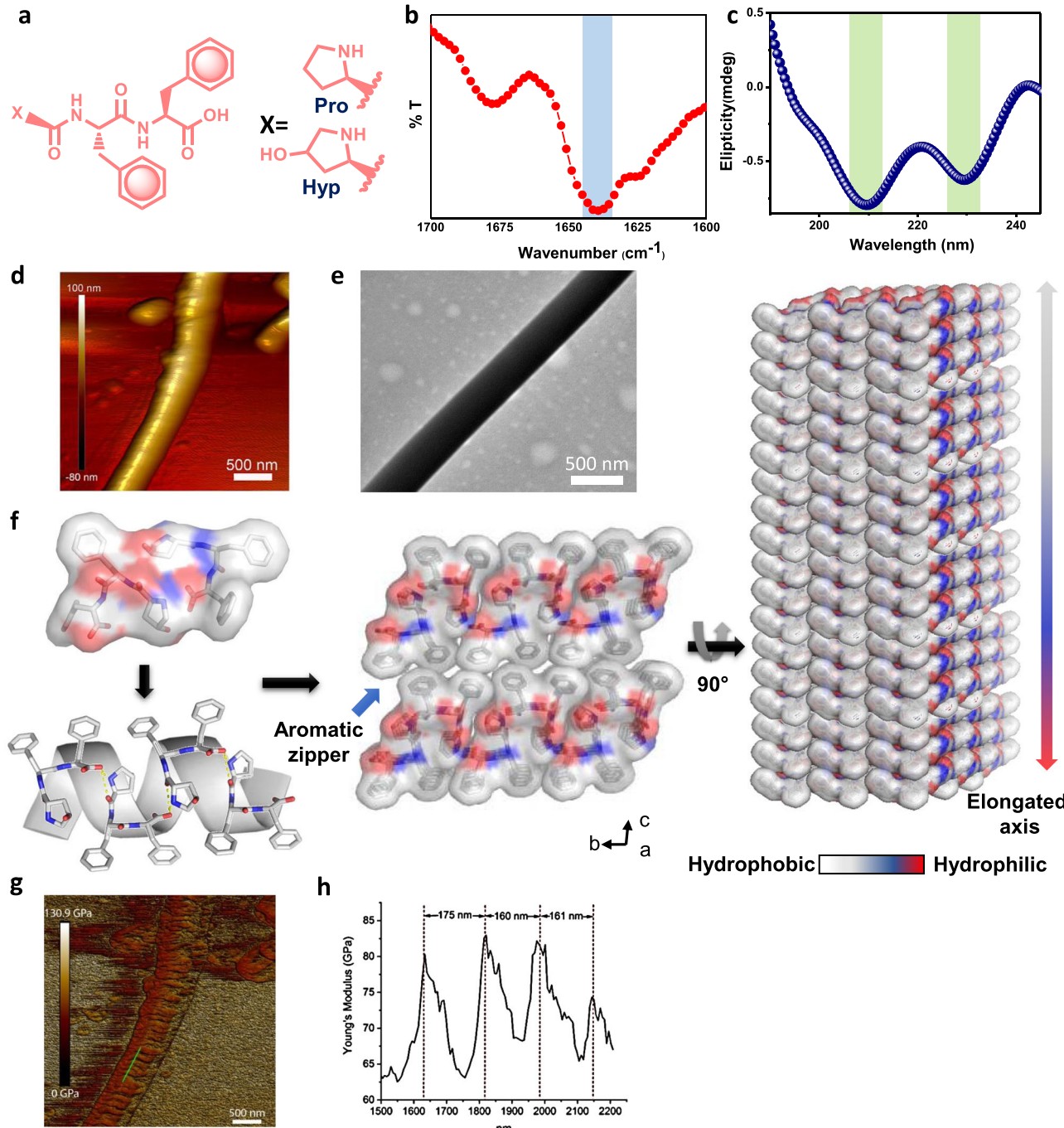

**Fig. 1 Self-assembly and structural characterization of Hyp-Phe-Phe. a** Chemical structure of Pro-Phe-Phe and Hyp-Phe-Phe. **b** FTIR analysis of Hyp-Phe-Phe assemblies showing the characteristic peak of helical conformation. **c** CD spectrum of the tripeptide reveals the presence of helical assemblies in solution. **d** AFM image of the tripeptide demonstrating fibrillar morphology. **e** TEM image of the tripeptide fibres. **f** Single-crystal structure of Hyp-Phe-Phe showing the formation of an elongated structure by stacking of helices through intermolecular hydrogen bonding and aromatic zipper-like packing. CCDC ref. no. 1823367[26]. **g, h** Mechanical strength of the Hyp-Phe-Phe fibres. **g** Nanoscale mapping of Young's modulus (Z-scale = 140 GPa). **h** Line section through one fibril as highlighted by the green line in **g**, showing the periodic variation in stiffness along the fibril. Additional AFM and TEM data are given in Supplementary Figs. 1–3.

assemblies. To investigate the nanomechanical properties of Hyp-Phe-Phe fibres, we used quantitative nanomechanical mapping AFM (QNM-AFM) (Fig. 1g, h). The measured Young's modulus of the fibrils varied in the range of 60–90 GPa, producing the same order of magnitude mechanical stiffness as recorded by AFM nanoindentation in the corresponding single crystal[26]. The preservation of the single-crystal level of mechanical rigidity in the peptide fibre was quite remarkable for a biomaterial

and comparable with stiff biological materials such as the bone, collagen and enamel[47] from which we conclude that the presence of the "aromatic zipper" molecular motif led to the fabrication of the stiff macroscopic biomaterial. Definite corrugation was observed perpendicular to the long axis of the fibril (green line in Fig. 1g), with a peak to peak distance of ~165 nm and a periodicity of 84.9 nm, as determined by direct Fourier transform analysis using the open source software FiberApp

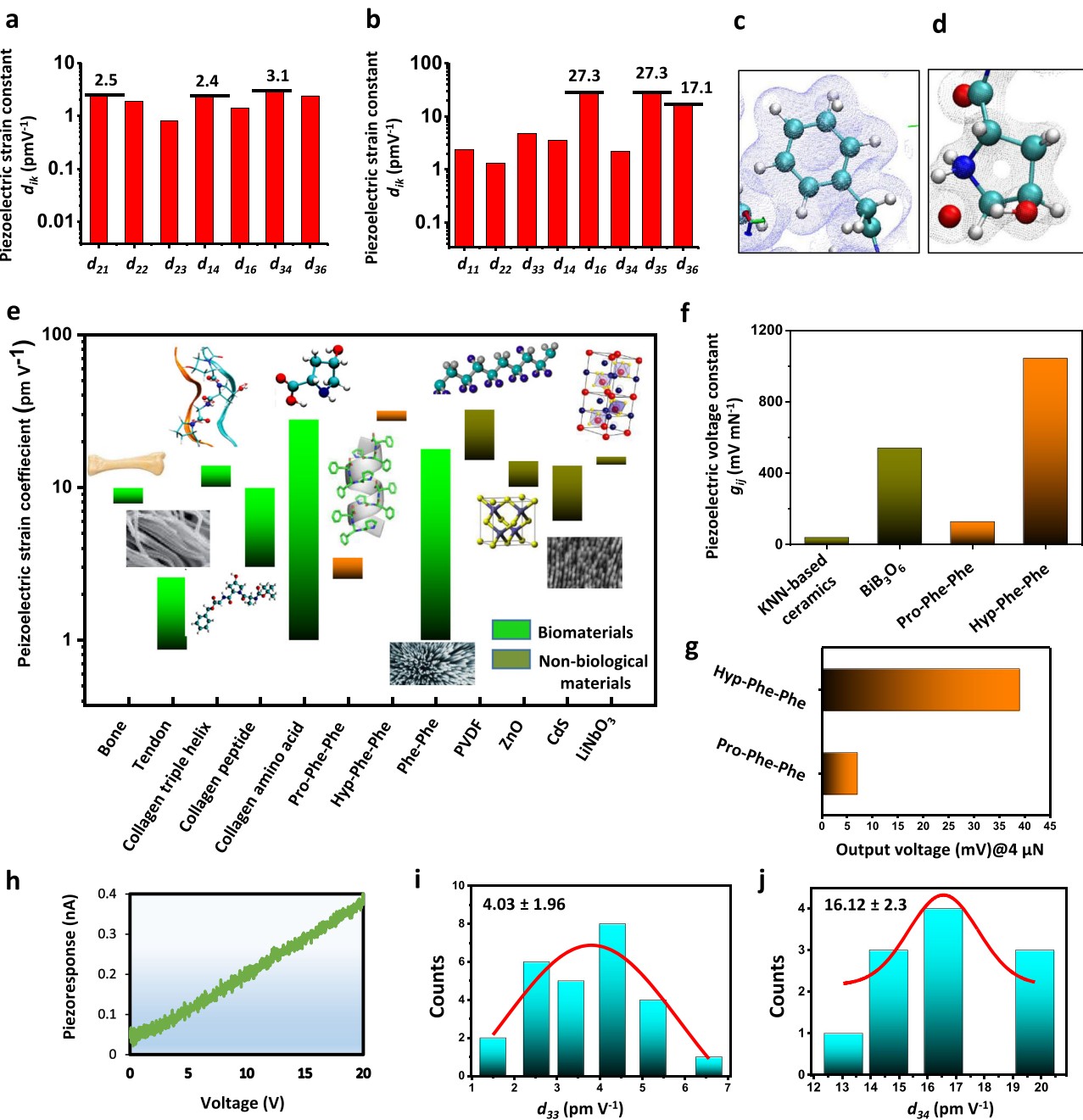

**Fig. 2 Piezoelectricity of Pro-Phe-Phe and Hyp-Phe-Phe assemblies. a**, **b** Calculated piezoelectric strain constants for Pro-Phe-Phe (**a**) and Hyp-Phe-Phe (**b**). Selected values are marked above relevant columns. **c**, **d** Zoom-in on the electronic structure of the Hyp-Phe-Phe crystal. The π-density of the diphenylalanine motif remains identical across both crystals (**c**) and **d** shows the hydroxylated pyrrolidine structure of the Hyp residue in the Hyp-Phe-Phe crystal. **e** Comparison of piezoelectric strain response of different biological and non-biological materials: collagen piezoelectric response at different length scales[10, 25, 53] has been emphasized as compared to that of Pro-Phe-Phe and Hyp-Phe-Phe. **f** The predicted voltage coefficient of Pro-Phe-Phe and Hyp-Phe-Phe in comparison with some currently used inorganic materials. **g** Calculated output voltage upon applied force of 4 μN. **h–j** Experimental measurement of coefficients using PFM. **h** Linear relationship between the vertical piezoresponse of Hyp-Phe-Phe as measured by the photodiode system and applied voltage. Statistical distribution of the vertical $d_{33}$ coefficients (**i**) and shear $d_{34}$ coefficients (**j**). Additional PFM data are given in Supplementary Figs. 5–16.

(Supplementary Fig. 3)[48]. This "mechanical periodicity" cannot be observed in the topography channel but is clear in the Young's modulus channel. Natural collagen exhibits similar lateral periodicity of 67 nm due to gaps in the hierarchical structure of the triple helix[49].

**Piezoelectric response of Pro-Phe-Phe and Hyp-Phe-Phe assemblies.** Having verified helical conformation, lateral periodicity and high mechanical robustness similar to collagen, we used

DFT to predict the elastic, dielectric and piezoelectric constants of the tripeptides (Fig. 2 and Table 1). Calculations were carried out on the obtained solid-state XRD crystal structures, which contained no waters of crystallization. Full details of the computational methodology can be found in the 'Methods' section. The Pro-Phe-Phe and Hyp-Phe-Phe assemblies have very similar predicted dielectric constants $\varepsilon_r$, with only a slight increase in the hydroxylated peptide (3.3 vs. 3.1, see Supplementary Table 2). This stems from 25% increase in the $\varepsilon_3$ tensor component for

**Table 1 DFT calculated molecular dipoles and crystal dipoles for each tripeptide, and the corresponding experimentally measured piezoelectric response.**

| Peptide | Molecule dipole (Debye) | Crystal dipole (Debye) | Space group no. | Longitudinal response permitted? | Shear response permitted? | Longitudinal response (pm V$^{-1}$) | Shear response (pm V$^{-1}$) |
|---|---|---|---|---|---|---|---|
| Pro-Phe-Phe | 7.9 | 2.8 | 4 | Yes | Yes | 2.2 | <0.1 |
| Hyp-Phe-Phe | 6.7 | 1.9 | 1 | Yes | Yes | 4.0 | 16 |

Hyp-Phe-Phe, compared to Pro-Phe-Phe (4.0 vs. 3.2). This axis of highest permittivity also gives the highest predicted piezoelectric charge tensor values. The lower shear elastic stiffness values in Hyp-Phe-Phe (Supplementary Table 3) indicate higher shear piezoelectric strain constants and voltage constants for this crystal, as the lower stiffness produces larger ionic displacement under an applied force. Supplementary Tables 4 and 5 show the DFT-computed piezoelectric charge ($e$), strain ($d$) and voltage ($g$) tensors of Pro-Phe-Phe and Hyp-Phe-Phe, respectively. For Pro-Phe-Phe, we observe that low charge tensor components of up to 56 mC/m$^2$ results in moderate $d_{ij}$ values of up to 3.1 pm V$^{-1}$ (Fig. 2a). However, as with most biomaterials[11], the low dielectric constants produce significant voltage constants of up to 108 mV m/N ($g_{22}$). In Hyp-Phe-Phe, hydroxylation lowers the symmetry of the unit cell (monoclinic to triclinic), which increases the number of non-zero piezoelectric constants in each tensor. Hydroxylation also gives a fivefold increase in the highest charge tensor value ($e_{33} = 0.1$ C m$^{-2}$). Due to the increase in $e_{ij}$ values and decrease in $c_{ij}$ values, we notice a significant increase in the magnitude of the predicted piezoelectric strain constants, with $d_{max} = d_{35} = -27.3$ pm V$^{-1}$ and $d_{33} = 4.8$ pm V$^{-1}$ (Fig. 2b). Looking at the full piezoelectric tensor of the crystals enables more meaningful comparisons to other piezoelectric materials and identifies possible applications for these crystals. The monoclinic Pro-Phe-Phe crystal has eight finite tensor components, whereas the low-symmetry Hyp-Phe-Phe crystal has 17 components. The most commonly exploited piezoelectric response in devices is the longitudinal $d_{33}$ value (or crystallographic equivalent $d_{11}$ and $d_{22}$). Both tripeptide crystals have modest longitudinal constants that would see them fit similar applications to aluminium nitride, quartz and zinc oxide (ZnO) as evidenced by the values cited in Supplementary Table 1. The Pro-Phe-Phe crystal shows a narrow range of tensor values, with coefficients in the range 0.8–2.5 pm V$^{-1}$. Hyp-Phe-Phe has a much wider range of values, varying from 0.1 pm V$^{-1}$ to the maximum 27.3 pm V$^{-1}$ ($d_{16}$ and $d_{35}$). The high predicted shear value and doubling of the longitudinal response in the Hyp-Phe-Phe crystal highlights that simple chemical modifications can induce piezoelectric response in biomaterials that exceeds that of many inorganic crystals (Fig. 2e). The maximum calculated response for Hyp-Phe-Phe is as large as the $d_{31}$ value of poled polyvinylidene difluoride, but is predicted to occur without the application of heat or a large external electric field. Given the similar $\varepsilon_r$ values for Hyp-Phe-Phe and Pro-Phe-Phe, we predict high voltage constants, on the order of 1 V m/N (Hyp-Phe-Phe, $g_{max} = g_{16} = 1043$ mV m/N). For comparison (Fig. 2f), (K$_{0.5}$Na$_{0.5}$)NbO$_3$ (KNN)-based ceramics have exhibited voltage constants of $g_{33} = 40$ mV m/N[31]. Other studies have reported values of up to 540 mV m/N ($g_{33}$) for single crystals of BiB$_3$O$_6$[32]. Combining our predicted piezoelectric voltage constants with approximate crystal dimensions, we can estimate single-crystal voltage outputs (Fig. 2g). For example, applying a loading force of 4 μN[50] along the two axis of a Pro-Phe-Phe single crystal will give a predicted voltage output of ~7 mV. For Hyp-Phe-Phe single crystal, the calculated voltage output is 39 mV, fivefold higher under similar conditions. The benefit of predicting the full piezoelectric tensor with DFT and screening for individually high-tensor components is that we can design devices and orientate biomolecular crystals and assemblies in a way that maximizes output.

The supramolecular packing modes in the crystal confer the significant $d_{33}$ response for Hyp-Phe-Phe, as applying a force along the 3-axis will align the hydroxylated pyrrolidines and couple to the net dipole in the unit cell. Our DFT and MD molecular models show that by strengthening the H-bonding network, we remodel the aromatic zipper motif and alter the net dipole in the unit cell (Table 1, Supplementary Table 6 and

Supplementary Fig. 4) to create more opportunities for charge transfer under stress. We have previously observed that lowering the crystal symmetry lowers the shear elastic constants and allows for more ionic displacement around the unit cell axes, for angles > 90° (in Hyp-Phe-Phe, the angle is 97°)[10]. Finally, the binding energy along the *3*-axis is slightly larger for Pro-Phe-Phe than for Hyp-Phe-Phe in the unstrained crystals (2.3 eV vs. 2.0 eV, Supplementary Table 7), which suggests that molecular arrangements that can facilitate an increase in intermolecular interactions under stress may create higher piezoelectric response.

To study the hierarchal influence of hydroxylation[51], we compare the current findings on tripeptide crystals and fibres to previous results obtained on Hyp and proline single crystals[25]. Only very small difference is observed in the predicted stiffness constants between proline and Hyp (24 GPa vs. 28 GPa), with this trend extending to the tripeptide crystals. The lower shear stiffness predicted for Hyp-Phe-Phe results in Young's modulus of 12 GPa, slightly below the value of 14 GPa for Pro-Phe-Phe. Both tripeptides and amino acids show large increase in charge tensor values after hydroxylation and decrease in stiffness constants, despite decrease in the predicted molecular dipole. The net crystal dipoles for both crystals are along the crystallographic *b* or piezoelectric *2* axis, with absolute values of 2.8 Debye for Pro-Phe-Phe and 1.9 Debye for Hyp-Phe-Phe. The projected dipole moments along each crystal axis (Supplementary Table 6) show that Pro-Phe-Phe has a unidirectional dipole moment, as it only has a *22* longitudinal response. The lowering of symmetry in Hyp-Phe-Phe gives three longitudinal components, which disperses the dipole moment along all three axes (Supplementary Fig. 4). The trend for piezoelectric strain tensor values in Pro-Phe-Phe and Hyp-Phe-Phe tracks almost exactly as that for proline and Hyp amino acids with an increase in $d_{max}$ from ~3 to ~30 pm V$^{-1}$.

To experimentally validate our predictive modelling, we first employed piezoresponse force microscopy (PFM) to characterize the crystals (Fig. 2h–j and Supplementary Figs. 5–16). Due to the small size of the peptide single crystals, conventional PFM imaging of topography, amplitude, phase, etc., was not possible. The crystals moved during imaging attempts. In line with best practice techniques[34,52], we then carried out PFM point measurements where the probe was brought into contact with the single crystal and held stationary while the applied voltage was varied and the piezoresponse recorded. These were used to generate plots similar to Fig. 2h, which were then used to create statistical distributions of the response. The measurements were carried out with the probe stationary relative to the sample and at a low frequency (21 kHz), which minimized artefacts resulting from topographic crosstalk or resonance enhancement of the signal. Stiff probes with a spring constant of 5–6 N m$^{-1}$ were used to mitigate electrostatic and flexoelectric contributions[34]. All measurements were carried out at 20 °C and 40% relative humidity (RH) ambient laboratory conditions to ensure uniformity in the measurement conditions. Identical point measurements were carried out on both positive and negative controls, to verify the accuracy of the technique and to rule out any instrumental backgrounds or parasitic effects contributing to the signal, as described in detail in Supporting Information section 3. The linear relationship between the piezoresponse as measured by the photodiode system and applied voltage, and also the minimal frequency dependence of output piezoresponse are indicative of a genuine piezoelectric property (Fig. 2h and Supplementary Figs. 13–16). The results revealed the vertical coefficient $d_{33}^{eff}$ of Pro-Phe-Phe assemblies to be 2.15 ± 0.86 pm V$^{-1}$ (Supplementary Fig. 13), which rose to 4.03 ± 1.96 pm V$^{-1}$ (Fig. 2i) for Hyp-Phe-Phe. Measuring the shear piezoelectricity of Hyp-Phe-Phe yielded an effective shear piezoelectric coefficient $d_{34}^{eff}$ of

16.12 ± 2.3 pm V$^{-1}$ (Fig. 2j), which is higher than the experimentally measured magnitude of LiNbO$_3$ (13 pm V$^{-1}$), ZnO (12 pm V$^{-1}$), amino acid γ-glycine (10 pm V$^{-1}$) and protein/peptide biomaterials M13 bacteriophage (6–8 pm V$^{-1}$) and collagen film (1 pm V$^{-1}$)[10,50,53].

To further probe the assemblies, we performed MD simulations (Fig. 3, Supplementary Figs. 17–22 and Supplementary Table 8). These simulations are performed on a nanocrystal of each peptide immersed in a large water box to model bulk solvation, as described in the 'Methods' section. MD are calculated at constant room temperature and atmospheric pressure at physiological pH of 7.4. The most important finding is that the tightening of the H-bond network in the Hyp variant is associated with increased conformational freedom in the Phe-Phe rings, showing the balance between electrostatic and π–π interactions in the creation of the helical assemblies. Figure 3 shows the computed supramolecular packing in both assemblies with the computed difference map in Fig. 3c comparing the separation vs. contact angle distributions in the Phe-Phe ring contacts for Hyp-Phe-Phe and Pro-Phe-Phe. The angle–distance contour plot makes it possible to identify different conformations between neighbouring phenyl rings, which helps characterize the changes induced by the additional –OH group in the Hyp variant. The difference map is used to highlight these changes. The negative value in the 5–6 Å and small angle region on Fig. 3c show that Pro-Phe-Phe presents larger density of highly ordered π–π contacts. By contrast, the Phe-Phe ring distance is shifted towards larger values in Hyp-Phe-Phe with an overall broadening of the density peaks, illustrating the increased conformational freedom. The evidence for the stability of the zipper for both Pro-Phe-Phe and Hyp-Phe-Phe have been obtained from both MD simulations and crystal structures, which confirm the rigidity and durability of the zipper as a structure-building motif (Supplementary section 4 and Supplementary Fig. 20–23).

**Characterisation of peptide-based power generator.** Finally, to test the potential of the peptide assemblies for energy harvesting and piezoelectric sensing in integrated microdevices, a coin-size power generator was designed and fabricated by tightly sandwiching the tripeptide assembly film between only the two Ag electrodes that were connected to the measuring instrument via copper wires (Fig. 4a and Supplementary Fig. 24). The entire device was firmly laminated with kapton tape to protect against mechanical stress, dust and humidity. When the device was compressed and released, the peptide-based nanogenerator converted the mechanical energy into electricity. Mechanical loads were applied to the generators using a dynamic mechanical test system and the resulting electrical output signal was characterized by measuring the short-circuit current and open-circuit voltage. A periodic compressive force was applied on the power generator and the output performance is shown in Fig. 4b, c. The switching-polarity test showed that the current and voltage signals were reversed when the device connection was switched (Fig. 4d, e)[54]. The switching-polarity test results excluded the errors from the variation of contact resistance or parasitic capacitance and confirmed that the detected electrical signal was truly from the piezoelectric peptide assemblies.

For Pro-Phe-Phe, under an applied force $F = 55$ N, the output open-circuit voltage ($V_{oc}$) reached 1.4 V (Supplementary Figs. 25a, c), which is significantly higher than reported peptide and inorganic alternatives (Supplementary Table 9). The corresponding short-circuit current ($I_{sc}$) was 52 nA (Supplementary Figs. 25b, d), which is significantly higher than the output current obtained from nanogenerators based on M13 bacteriophage virus (6 nA) or fish skin collagen (1.5–20 nA) (Supplementary Table 9).

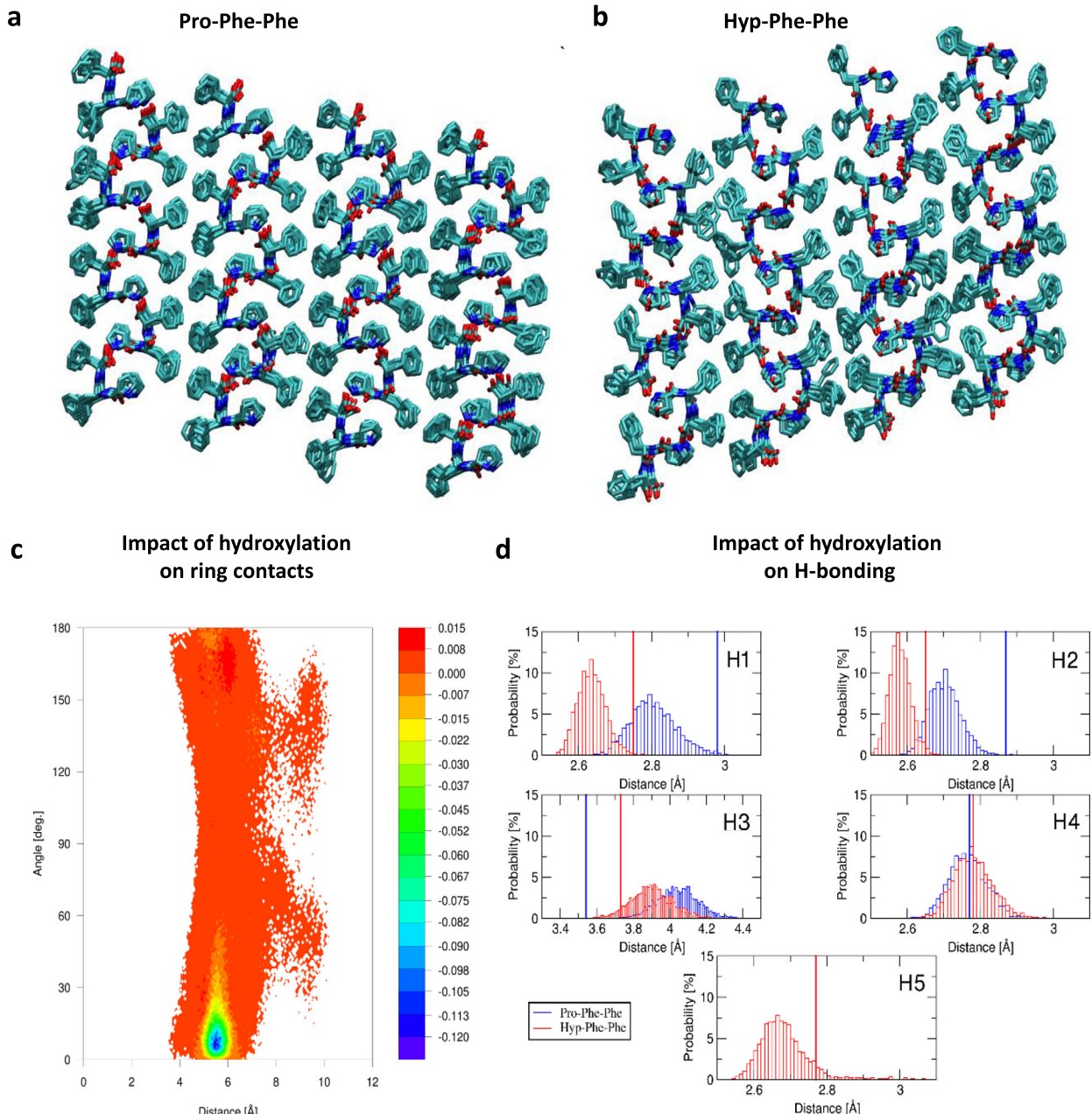

**Fig. 3 Molecular dynamics (MD) simulations of Pro-Phe-Phe and Hyp-Phe-Phe assemblies. a, b** The computed supramolecular packing structure in the assemblies after 0.1 μs of room temperature dynamics of Pro-Phe-Phe (**a**) and Hyp-Phe-Phe (**b**). The balance between increased H-bonding and more flexible π-π contacts is quantified in the Pro-Phe-Phe vs. Hyp-Phe-Phe difference map in **c**. The distance corresponds to the distance between neighbouring phenyl rings and the angle to the angle between the same two rings, as described in the 'Methods' section. The density is the normalized histogram and is given as a percentage. The 2D histogram is built with a 200 × 200 matrix. H-bond distributions are plotted in **d**. Additional supporting MD data are provided in Supplementary Figs. 14–19 and Supplementary Table 8.

Using the Hyp-Phe-Phe assemblies as the active layer (Fig. 4f, g), similar short-circuit current output (39.3 nA) was achieved when the applied force was only 23 N, half that applied for Pro-Phe-Phe. This validated our theoretical predication of increased piezoelectric response due to simple hydroxylation of the side chain. The corresponding output voltage was 0.45 V. The peak voltage and current increased linearly as a function of the applied force at rates of 15.08 mV N$^{-1}$ and 1.33 nA N$^{-1}$, respectively (Fig. 4h, i), demonstrating linear piezoelectric response of the peptide assemblies. Furthermore, the high mechanical rigidity suggests that power generation can be sustained under a cyclic force (17 N) (Supplementary Fig. 26) and the output voltage showed no

degradation over 1000 press/release cycles for more than 60 min, indicating the high durability of the peptide-based devices. Finally, the voltage output characteristic was measured with different external load resistors connected to a nanogenerator, while it was repeatedly deformed, and the result is shown in Supplementary Fig. 27. The output voltage continually rises with the growth of load resistance, demonstrating the electrical characteristics of the power device and illustrating its potential for practical applications.

We expect that the device performance can be further increased in the future by fabricating highly ordered aligned arrays[42], which will allow to scale up our peptide-based piezoelectric devices to generate a higher energy output. To

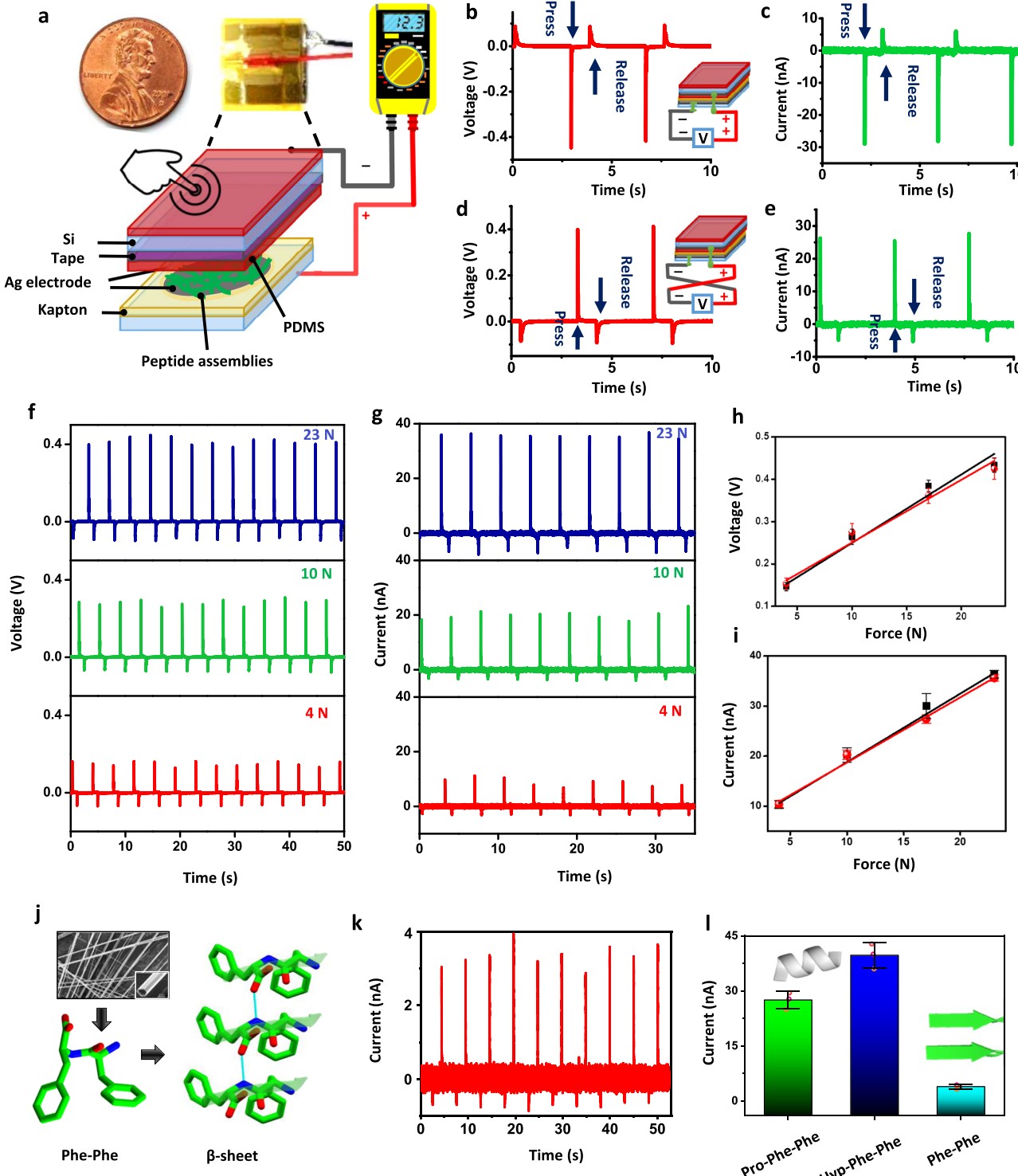

**Fig. 4 Characterization of the helical peptide-based nanogenerator. a** Photograph and schematic configuration of coin-sized piezoelectric energy generation measurement set-up utilized as a direct power source with peptide assemblies as the active component. **b, c** Open-circuit voltage (**b**) and short-circuit current (**c**) of piezoelectric energy harvester in the forward connection. **d, e** The generated voltage output (**d**) and current output (**e**) in the reverse connection. **f, g** Variable external load was applied onto the device. The measured open-circuit voltage (**f**) and short-circuit current output (**g**) of Hyp-Phe-Phe under different applied force. **h, i** Linear dependence of the voltage (**h**) and current (**i**) output on the applied force. The black and red lines indicate forward and reverse connection, respectively. **j–l** Comparison of piezoelectric response of the engineered helical tripeptides with β-sheet forming Phe-Phe peptide assemblies. **j** The self-assembled nanostructures and β-sheet molecular arrangement in the crystal structure of Phe-Phe (CCDC ref. no 16340[83]). **k** Short circuit current obtained from the generator using Phe-Phe assemblies as the active component upon applied force = 23 N. **l** Comparison of output current of Phe-Phe with helical peptides, Pro-Phe-Phe and Hyp-Phe-Phe. Error bars = SD (n = 3).

further understand the relation between atomic level structural organization and macro-scale piezoelectricity, we fabricated a control device using the well-established β-sheet forming dipeptide, Phe-Phe (Fig. 4j and Supplementary Figs. 28 and 29). Under the same applied force, $F = 23$ N, the maximum output open-circuit voltage and short-circuit current was only 0.14 V (Supplementary Fig. 29a, b) and 3.9 nA (Fig. 4k), respectively. Although current and voltage output increased linearly with mechanical force (Supplementary Fig. 29c, d), the value was much lower compared to the helical peptides even at the highest applied force (Fig. 4l). The significantly decreased performance of the analogous aromatic dipeptide under a similar applied force highlights how differences in the structural organization at the atomic level can bear a dramatic effect on biomaterial piezoelectricity.

**Structural component, molecular organization and piezoelectric response**. Currently, the piezoelectricity of biomaterials is not fully understood at the molecular level. The hierarchical structural organization of biomaterials is not included in the classical piezoelectric models, leading to discrepancies between their predictions and experimental results[55]. There is emerging evidence that the piezoelectric response of biomaterials profoundly depends on the atomic level structure and self-assembled hierarchical organization, as well as on the amplitude and sign of the dipoles of the constituent amino acid residues. Both collagen and its tripeptide minimalistic analogues Pro-Phe-Phe and Hyp-Phe-Phe exhibit helical structures. Moreover, the shortest repeating unit of collagen also displays a molecular arrangement and symmetry similar to that of the studied tripeptides[56]. The Phe groups have been assumed to facilitate aromatic molecular orbital overlap in the underlying parent peptide, potentially leading to electron delocalization and generation of a significant band gap[28]. Thus, previous molecular design incorporated aromatic Phe residues in helical peptide sequences in order to engineer electron transport properties in the resulting assembled structures. This is consistent with the lower piezoelectric response of the Hyp-Leu-Phe peptides, which retains the helical organization in the crystal but possesses reduced aromaticity (Supplementary section 6). At the same time, among the collagen amino acids, Hyp showed the highest piezoelectric response in a single-crystal form[25]. Our molecular models show that hydroxylation significantly increases the crystal polarizability under stress. Thus, Hyp-Phe-Phe satisfies all the specific requirement of high piezo sensitivity in terms of structure and interactions, thereby exhibiting the highest piezoelectric response among the studied natural short peptides.

Exploring the potential of renewable, sustainable and green energy sources to replace fossil fuels is one of the most significant and urgent challenges in energy research. The piezoelectric effect in proteins is an intriguing phenomenon that can potentially allow a better interface between the semiconductor and biological worlds. Molecular engineering of peptide piezoelectricity is critical not only to understand the molecular basis of piezoelectricity in biomaterials, but also to unravel the core recognition motif for modular design of short peptides with a predictable high piezoelectric response. Here we fabricated a simple biopiezoelectric device made from collagen-mimicking ultra-short peptide sequences that could achieve high current and voltage output, similar to that obtained using nanogenerators comprising inorganic materials or organic polymers. We found common helical molecular organization in the packing of collagen and our designed short peptides, which can explain their high piezoelectricity. A significantly enhanced energy output can be obtained by simple replacement of one constituent with another of higher polarizability (in our case, C-H → C-OH on a

pyrrolidine ring). This engineered higher response to applied strain emphasizes the importance of incorporating polar building units in piezoelectric biomaterials. Moreover, sequence mutation by replacing Phe for Leu noticeably decreases the piezoelectric response, although having similar molecular organization emphasizing the significant role of aromatic groups. Our demonstration of collagen-level piezoelectricity in rationally designed ultra-short peptides reinforces the value of prediction-led molecular engineering of piezoelectricity to accelerate the deployment of peptides in nanotechnology applications.

## Methods

**Preparation of peptide assemblies**. The peptides were purchased from DGpeptides Co., Ltd (Hangzhou City, Zhejiang Province, China) and purified to >95%, and their identity was confirmed by mass spectrometry. All peptides were stored at −20 °C. For assembly, peptides were dissolved in phosphate buffer at pH 7.4 to a final concentration of 1.5 mg ml$^{-1}$ by heating at 90 °C and vigorous vortexing for 2 min. The peptide solutions were then incubated at 18 °C for 2 weeks with frequent shaking before examination.

**FTIR spectroscopy**. A 30 μl aliquot of the peptide solution was deposited onto disposable KBr infrared sample cards (Sigma-Aldrich, Rehovot, Israel), which were then allowed to dry under open air and followed by vacuum. The samples were saturated twice with 30 μl of D$_2$O and vacuum dried. FTIR spectra were collected using a nitrogen-purged Nicolet Nexus 470 FTIR spectrometer (Nicolet, Offenbach, Germany) equipped with a deuterated triglycine sulfate detector. Measurements were performed at 4 cm$^{-1}$ resolution and obtained values were averaged over 32 scans. The absorbance maxima values were determined using an OMNIC analysis programme (Thermo Scientific Nicolet). The background was subtracted using a control spectrum.

**CD spectroscopy**. CD spectra were collected using a Chirascan spectrometer (Applied Photophysics, Leatherhead, UK) fitted with a Peltier temperature controller set to 25 °C, using quartz cuvettes with an optical path length of 0.1 mm (Hellma Analytics, Müllheim, Germany). Absorbance of the sample was kept within the linear range of the instrument during measurement. Data acquisition was performed in steps of 1 nm at a wavelength range of 190–240 nm with a spectral bandwidth of 1.0 nm and an averaging time of 3 s. The spectrum of each sample was collected three times and averaged. Baseline was similarly recorded for phosphate buffer and subtracted from the samples spectra. Data processing was performed using Pro-Data Viewer software (Applied Photophysics, Leatherhead, UK).

**Transmission electron microscopy**. A 5 μl aliquot of the peptide solution was placed on 400 mesh (theoretical square size of ~64 μm) copper grids. After 1 min, excess fluids were removed. For negative staining, the grid was stained with 2% uranyl acetate in water and after 2 min, excess fluids were removed from the grid. TEM micrographs were recorded using a JEM-1400 (JEOL, Tokyo, Japan) operating at 80 kV.

**Quantitative nanomechanical AFM**. Hyp-Phe-Phe fibres were formed by dissolving the peptide (1.5 mg ml$^{-1}$) in Milli-Q® water at 90 °C for a minimum of 3 h to ensure the complete solvation of the monomeric peptide. During the course of the heating, the mixture was vortexed vigorously once every hour. After heating, the samples were cooled slowly and self-assembly was allowed to proceed over a period of 1 week at room temperature. All QNM-AFM experiments were performed in peakforce mode on a Multimode 8 AFM (Bruker) and a Nanoscope V controller (Bruker). All imaging was performed in air using RTESPA 525 cantilevers with a spring constant of ~200 N m$^{-1}$ and a resonant frequency of 525 kHz were used.

**Density functional theory**. Periodic DFT calculations[57] were performed on the single crystals using the VASP[58] code. Electronic structures are calculated using the Perdew–Burke–Ernzerh (PBE) functional[59], projector augmented wave (PAW) pseudopotentials[60] with a plane wave cut-off of 1000 eV, and k-point sampling of $4 \times 4 \times 4$. A finite differences method was used to calculate the stiffness tensor, with each atom displaced in each direction by ±0.01 Å, using a plane wave cut-off of 1000 eV, and k-point sampling of $2 \times 2 \times 2$. Piezoelectric strain constants and dielectric tensors were calculated using density functional perturbation theory[61] with a plane wave cut-off of 1000 eV, and k-point sampling of $2 \times 2 \times 2$. Binding energies were calculated between the two molecules in each unit cell using Grimme-D3 dispersion corrections[62].

Single-molecule dipoles were calculated from gas-phase molecule electronic structures solved using the Gaussian16 software package[63], with the hybrid HF-DFT B3LYP functional[64,65] and 6-311 + +g** basis set. Crystal dipole moments were calculated using periodic DFT[57] with the CP2K[66] code. Electronic structures are calculated using the PBE functional[59] and PAW pseudopotentials[60] with a

combination of plane wave and Gaussian basis sets[67,68]. The combination is performed on a multigrid with five levels of decreasing resolution. The cut-off of the finest grid was 1300 Ry (~17687 eV) and the cut-off controlling the grid at which a Gaussian is mapped was 70 Ry (~952 eV). The basis set was a double-zeta with polarization orbitals[69]. The electronic cycle was performed with the orbital transformation[70] (OT) method, which uses the derivative of the wave function to reach the minimum of energy. The combination of a localized basis set with OT makes it possible to obtain the crystal dipole moment using the Berry phase approach. A supercell of $3 \times 2 \times 2$ unit cell units was used and both the lattice parameters and the ions were free to move during the geometry optimization. The space group symmetry is imposed through a homemade patch to CP2K.

**Molecular dynamics**. MD simulations were performed with the Gromacs 2018.4 package. All bonds involving a hydrogen atom were constrained using the LINCS[71] algorithm, which allowed the use of an integration timestep of 2 fs in the leapfrog integrator[72], and coordinates were saved every 10 ps. Long-range electrostatics were treated by the Particle Mesh Ewald method[73].

All systems were minimized for 100,000 steps and heated progressively from 0 to 300 K for a total 3 ns at constant volume, followed by 100 ns of equilibration at constant pressure and temperature. During equilibration, the reference temperature was set at 300 K with a time constant of 1 ps and the reference pressure was set at 1 bar with a time constant of 5 ps, using the Berendsen[74] thermostat and barostat, respectively. The subsequent production phase of dynamics was carried out in the constant pressure and temperature NPT (isothermal–isobaric) ensemble, with the reference pressure set at 1 bar with a time constant of 5 ps using the highly accurate Parrinello–Rahman barostat[75], and all molecules were coupled separately in groups to an external heat bath set at 300 K with a coupling time constant of 1 ps using the velocity-rescale thermostat[76]. The pH is set by the protonation state of the amino acids at the physiological pH of 7.4 used in the crystallization experiments.

VMD 1.9.3 software was used to generate supercells from the crystal unit cell[77]. Cells were replicated $14 \times 10 \times 6$ times in the $x$, $y$ and $z$ directions, respectively. Supercells of ~$7.1 \times 9.5 \times 12$ nm in dimension were then placed in large water boxes and subjected to room temperature constant-pressure MD using the GROMACS code with the CHARMM 36 m forcefield and TIP3P water[78]. Hyp is not included in the CHARMM36m forcefield, so we used the Swissparam server to generate its parameters[79]. Structures were heated to room temperature from 0 K over 5 ns of constant volume dynamics and then subjected to 100 ns of dynamics at constant pressure. In the core of the supercells, hydrogen bond statistics were measured across >10,000 structures during the final 50 ns of dynamics using VMD tools and in-house python scripts, together with contact distances between the stacked phenyl rings and between neighbouring helical structures for T-stacking rings. The distance between neighbouring rings is taken as the distance between their centres of mass as estimated by the midpoint between the γ-carbon and the two carbon atoms at meta positions on the phenyl ring. These three carbons are also used to define the plane of the ring, with the angle between two rings formed by the normal to each ring plane.

To validate the use of the CHARMM 36 m forcefield, designed for globular proteins in solution, and the set of parameters provided by SwissParam to model tripeptide crystals, the stability of the peptide crystal structures was tested using CP2K[65] with the supercell approximation. For both Pro-Phe-Phe and Hyp-Phe-Phe tripeptides, the supercell used was $7 \times 3 \times 2$ (the numbers indicate the number of replicas of the unit cell along the corresponding lattice vector). A cut-off of 1.2 nm, the same that was used for the MD simulations, was used for intermolecular interactions and the Ewald sum was used for long-range electrostatic interactions.

The unit cell space group and translation symmetries are conserved in the DFT calculations used to compute the electromechanical properties of the single crystals. They are not conserved in the MD simulations, as the crystal supercell or block is immersed in a large box of pre-equilibrated bulk water. These MD simulations were used to monitor the evolution of the supramolecular $\pi$–$\pi$ and H-bond contacts in the tripeptide assemblies in the central region of the block, to obtain time average values and error bars for the contacts identified in the starting X-ray structure.

**Fabrication of peptide-based power generator**. Two $0.7 \times 0.7$ cm$^2$ Ag layers were deposited on two $1.2 \times 1.2$ cm$^2$ silicon substrates that served as the top electrode and the bottom electrode. The smooth surface of a Kapton tape was first coated with a polydimethylsiloxane (PDMS) layer of thickness 0.40 mm. A strip of double-sided tape was then attached to the top surface of PDMS. A $0.7 \times 0.7$ cm$^2$ hole was cut through the Kapton/PDMS/double-sided tape structure. The Kapton tape was attached to one silicon substrate with the Ag electrode exposed to the hole. The hole was then filled densely with the Phe-Phe, Pro-Phe-Phe and Hyp-Phe-Phe peptide assemblies, respectively. The other silicon substrate sputtered with silver film was then attached to the double-sided tape to complete the device fabrication without leaving any gap in the nanogenerator. To prevent cracking of the substrate due to the impact force from the linear motor, the top substrate was drop-coated with a PDMS damping layer. Two copper wires were connected to the silver electrodes using carbon ink (JELCON CH-8 MOD2) to complete the device fabrication. The stacking pattern of layers of the device is completely different from triboelectric-based nanogenerators[80,81].

**Characterization of the power generator**. The peptide-based power generator was vertically fixed onto a stainless-steel plate, which was mounted on a precision linear slide stage. A linear motor (E1100-RS-HC type with Force Control, LinMot) was used to periodically press the power generator. The outputs of the generator were recorded using a low-noise voltage preamplifier (Stanford SR560). The generator and measuring instruments were placed in a Faraday cage to avoid interference from the environment, and the linear motor was placed outside of the cage.

**Crystal preparation and data collection**. Crystals used for data collection were grown using the vapour diffusion method. The dry peptide was first dissolved in water, at a concentration of 5 mg/ml. Then, 50 μl was deposited into a series of $8 \times 40$ mm vessels. Each tube was sealed with Parafilm®, in which a single small hole was pricked using a needle. The samples were placed inside a larger vessel filled with 2 ml of acetonitrile. The systems were ultimately capped and incubated at 18 °C for several days. Needle-like crystals grew within 10–12 days. For data collection, crystals were coated in paratone oil (Hampton Research), mounted on a MiTeGen cryo-loop and flash frozen in liquid nitrogen. Diffraction data were collected at 100 K on a Rigaku XtaLab$^{Pro}$ with a Dectris Pilatus R 200K-A detector using Mo radiation $\lambda = 0.71073$ Å.

**Processing and structural refinement of crystal data**. The diffraction data were processed using CrysAlisPro 1.171.39.22a. Structure was solved by direct methods in SHELXT-2014/5[82]. The refinements were performed with SHELXL-2016/4 and weighted full-matrix least-squares against $|F|$ using all data. Atoms were refined independently and anisotropically, with the exception of hydrogen atoms, which were placed in calculated positions and refined in a riding mode. Crystal data collection and refinement parameters are shown in Supplementary Table 10, and the complete data can be found in the cif file as supplementary information. The crystallographic data have been deposited in the CCDC with no. 2008471 for Hyp-Leu-Phe.

## Data availability
Additional data that support the findings of this study are available from the corresponding authors upon reasonable request. The X-ray crystallographic coordinates for the atomic structure reported in this study have been deposited at the Cambridge Crystallographic Data Centre (CCDC) under deposition number CCDC 2008471 for Hyp-Leu-Phe. This data can be obtained free of charge from the Cambridge Crystallographic Data Centre. Source data are provided with this paper.

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

## Acknowledgements

S.B. thanks Tel Aviv University for post-doctoral fellowship. This project received funding from ERC under the European Union Horizon 2020 Research, innovation programme (grant agreement number BISON-694426 to E.G.) and the National Natural Science Foundation of China (NSFC No. 51973170 to R.Y.). D.T. thanks Science Foundation Ireland (SFI) for support under awards number 15/CDA/3491 and 12/RC/2275 and for supercomputing resources at the SFI/Higher Education Authority Irish Center for High-End Computing (ICHEC). We thank Dr. Sigal Rencus-Lazar for help in scientific and language editing.

## Author contributions

S.B., S.G. and H.Y. contributed equally to this work. S.B. and E.G. designed the experiments. S.B. performed the experiments and crystallization of peptides. S.G., J.O'D., O.M., P.A.C., S.A. M.T. and D.T. investigated the piezoelectric coefficients. H.Y. and R.Y. fabricated the power generators and evaluated their performance. N.P.R. measured the mechanical properties of the fibres. W.J. performed the TEM of the peptide. L.J.W.S. solved the crystal structure. S.B. coordinated all the work and analysed the results. S.B., S.G., D.T. and E.G. wrote the paper. All the authors commented on the manuscript.

## Competing interests

The authors declare no competing interests.
