## [Peer Review File · Nature Communications]

REVIEWER COMMENTS

Reviewer #1 (Remarks to the Author):

Molecular Engineering of Piezoelectricity in Collagen-Mimicking Peptide Assemblies

This manuscript demonstrates a peptide-based piezoelectric generator using a helically arranged peptides with high piezoelectricity. The authors claim that the simple addition of a hydroxyl group increased the piezoelectric response by an order of magnitude ($d_{35}=27 \text{ pC N}^{-1}$), which is similar to that of the highest predicted to date in short natural peptides. Thus, the authors provide promising device demonstration of computationally guided molecular engineering of piezoelectricity in peptide nanotechnology and biocompatible application. However, the following are some questions for the authors to consider further improve the quality of the manuscript. First, in the view of the 'high piezoelectricity', the authors provided piezoelectric coefficients of several conventional materials and biomaterials to compare their piezoelectricity. However, there are researches on higher piezoelectric coefficients of the referred materials by diverse engineering methods. Therefore, the authors should compare piezoelectricity with the enhanced piezoelectric property of the materials. Also, how can the authors claim the piezoelectricity of the materials without comparing the various piezoelectric coefficients (d_{11} , d_{13} , d_{33} , etc.) of the materials? Can authors still claim significantly higher piezoelectricity than other materials?

Second, the computational works provide sufficient explanations and supplement the studies. Therefore, the modeling of the molecular structure of the material and the simulating conditions of the surroundings are crucial. What is the pH condition? The authors should provide more information about the experimental methods because the dipole or polarization of the material will be easily affected by the pH conditions. Also, how did the authors defined the surrounding conditions regarding the water contents?

The water contents (humidity) may strongly affect the piezoelectric property, molecular structure, and further mechanical strength of the materials. Moreover, how was the molecular structure of the peptides optimized? This is because the simulation works may show different results depending on the optimized structure of the materials. Therefore, more detailed explanations of how simulation was performed will help the reader understand and improve the quality of the article as well.

Third, from the last sentence of page 5 to the first sentence of page 6, what is the solid evidence in that the neighboring helical structural modules connected in a parallel orientation with the interface of the dimer were stabilized by the aromatic zipper structure built from pi-pi interactions between the Phe side chains. Since this is crucial to explain the extended helical-like molecular organization of the material, simply providing an assumption by referring other studies to explain the authors' theory seems less convincing. Therefore, the authors should provide solid evidence for the explanations.

Fourth, for the PFM measurements, the authors are only providing the piezoelectric coefficients of different peptide assemblies without any experimental images including topography, amplitude, deflection, and phase. These detailed experimental results should be supplemented to rule out the artifact-induced piezoelectric property of the materials.

Fifth, for the characterization of the peptide-based power generator, the authors should provide more detailed information about the device and the characterization results. In Figure 4a, the schematic image shows that the peptide assemblies are deposited on Kapton, and PDMS is also included between the electrodes. How can the authors be sure that the peptide assemblies combined with Kapton and PDMS show the pure piezoelectric property of the power generator? These materials may affect not only the electrical characterization, but also to the effective force application to the device. Therefore, the applied force indicated in Figures 4h and 4i may not be the effective force that was applied to the device. Moreover, the authors should also provide the voltage and current based on the different resistances to see the electrical characteristics of the power device. Furthermore, since the authors are also emphasizing the high mechanical property of the material, the durability test of the power generator should be supplemented.

Sixth, the use of expressions such as "quite, slight, slightly, very small, similar, significantly, only, only very, etc." leads to an unclear and ambiguous interpretation of the piezoelectricity of the materials. When making comparisons, the authors should clearly deliver the meanings by presenting numerical values or only large-small comparisons.

Reviewer #2 (Remarks to the Author):

This manuscript reports two Phe-Phe-derived peptide crystals that demonstrate high piezoelectricity. Using PFM, together with DFT and MD simulations, the authors showed that simple sidechain engineering of tripeptides, such as adding a hydroxyl group, could dramatically increase peptide crystals' piezoelectric response. The authors also demonstrated a large scale tripeptide-based generator that can produce current and voltage. While reading the manuscript, I was impressed by these tripeptides' extremely high piezoelectric constants. For example, the piezoelectric voltage constant of Hyp-Phe-Phe is even higher than that of widely used PZT. However, I have some concerns related to the origin of their measured electromechanical response using PFM and the large scale peptide generator, since other mechanisms could also lead to piezoelectric-like behaviors. Here, I provide comments that could be used to verify tripeptides' piezoelectric properties and to further improve the manuscript.

There have been many discussions in the field that PFM may not be a reliable tool to quantify nanomaterials' piezoelectricity (Gruverman, Nature Communications 10, 1661 (2019)), because many non-piezoelectric materials, such as HfO₂ (Kim, ACS Nano 6, 7026 (2012); Blake, ACS Nano 9, 6484 (2015)), can mimic piezoelectric behaviors in PFM measurements, and it is challenge to characterize the true piezoelectric constants due to local electrostatic effects and non-uniform electric fields. Also, the measurement could be very sensitive to local environmental conditions, i.e. temperatures and humidity. Clearly, the only voltage vs. displacement PFM results presented in the paper is not sufficient to proof peptides' piezoelectricity. The authors should perform in-depth experiments and data interpretations to verify and quantify the piezoelectric effects. For example, as suggested in Blake, ACS Nano 9, 6484 (2015), showing hysteresis loops of PFM amplitudes and phases with various electric fields could dramatically improve the accuracy of measured piezoelectricity. In addition, the authors should provide more information: If the measured sample is a single crystal or not? If not, how to compare the PFM data to simulation results? Do these peptides need any polarization to exhibit piezoelectricity, and why?

For the peptide-based nanogenerator, it might not be appropriate to claim that the measured voltage and current outputs originate from the piezoelectric effect, and switching the connect is not sufficient to exclude other electromechanical effects. For example, triboelectric generators with similar sandwiched electrode/PDMS/electrode or electrode/PDMS composite/electrode structures can produce equivalent or much higher voltages and currents with similar patterns, such as that demonstrated in Chen. ACS Applied Materials & Interfaces 8, 736 (2016) and He, Nano Research 9, 3714 (2016). Given that tripeptides could exhibit extremely high voltage constants, the authors should have a rational explanation about why the measured voltage output from the nanogenerator is relatively low. In addition, measuring the power output with various external loads will better illustrate its potential for practical applications.

Minor comments:

Page 6-7: The QNM results could be largely affected by peptide fibres' surface geometry, and it is very sensitive to topography changes. For example, a smaller contact area between the AFM tip and the sample will lead to a higher measured Young's modulus. The authors should be more careful about the claims in Fig 1g-I, and tip/surface contacts should be analyzed.

Page 6: Fig. 1g is confusing. The substrate image is identical to its topography image in Fig. 1d, but the legend only shows Young's modulus.

Page 6: Add a legend in Fig. 1i.

Page 10: The authors could uniform the units of the calculated dik constants in Fig. 2a-b and the measured ones in Fig. i-j.

Page 10: Fig. 2h can be changed to electric field vs. strain.

SI section: Fig. S1bc need to show their substrates

Author Response to the reviewer comments for “Molecular Engineering of Piezoelectricity in Collagen-Mimicking Peptide Assemblies”

Reviewer 1

Reviewer’s opening remarks:

This manuscript demonstrates a peptide-based piezoelectric generator using a helically arranged peptides with high piezoelectricity. The authors claim that the simple addition of a hydroxyl group increased the piezoelectric response by an order of magnitude ($d_{35}=27 \text{ pC N}^{-1}$), which is similar to that of the highest predicted to date in short natural peptides. Thus, the authors provide promising device demonstration of computationally guided molecular engineering of piezoelectricity in peptide nanotechnology and biocompatible application. However, the following are some questions for the authors to consider further improve the quality of the manuscript.

Authors’ response:

We thank the referee for their positive comments on the quality of our work and for acknowledging its potential impact on the field of rationally-designed biocompatible device materials. We have made extensive revisions and additions to our manuscript based on the detailed and stimulating comments, which we are very grateful for and which have helped us create a more wide-ranging and useful work.

1. Reviewer’s comment:

First, in the view of the ‘high piezoelectricity’, the authors provided piezoelectric coefficients of several conventional materials and biomaterials to compare their piezoelectricity. However, there are researches on higher piezoelectric coefficients of the referred materials by diverse engineering methods. Therefore, the authors should compare piezoelectricity with the enhanced piezoelectric

property of the materials. Also, how can the authors claim the piezoelectricity of the materials without comparing the various piezoelectric coefficients (d_{11} , d_{13} , d_{33} , etc.) of the materials? Can authors still claim significantly higher piezoelectricity than other materials?

Authors' response:

We thank the reviewer for raising this point. We agree with the referee that the comparison should be made with the best reliable reports in the literature. We have now significantly expanded **Supplementary Table 1** to include a broader range of piezoelectrics and ensured that the numbers we cite are the highest obtained by diverse engineering methods.

Regarding the second point above, when comparing piezoelectric responses of various materials, we usually compare first the largest component of the tensor, and then, depending on the experimental measuring technique, we compare the largest components that are distinguishable using that particular method. We appreciate and agree with the reviewer's concern that a material for which all components of the piezoelectric tensor are high may be claimed as having a higher net response than another material with just a single very high component. This is a very relevant question from an engineering point of view, and we have clarified that screening for individually high tensor components allows us to design devices and orientate biomolecular crystals and assemblies in a way that maximises output. We have added further DFT discussion on the range of piezoelectric constants in both of our collagen-mimicking peptide materials, and we have clarified which tensor components we refer to when comparing to other piezomaterials both in the main text and in Supplementary Table 1.

Changes made to the manuscript:

Page 9, lines 4-18:

“Looking at the full piezoelectric tensor of the crystals enables more meaningful comparisons to other piezoelectric materials and identifies possible applications for these crystals. The monoclinic Pro-Phe-Phe crystal has 8 finite tensor components, whereas the low symmetry Hyp-Phe-Phe crystal has 17 components. The most commonly exploited piezoelectric response in devices is the longitudinal d_{33} value (or crystallographic equivalent d_{11} and d_{22}). Both tripeptide crystals have

modest longitudinal constants that would see them fit similar applications to aluminium nitride (AlN), quartz and zinc oxide (ZnO) as evidenced by the values cited in Supplementary Table 1. The Pro-Phe-Phe crystal shows a narrow range of tensor values, with coefficients in the range 0.8–2.5 pm/V. Hyp-Phe-Phe has a much wider range of values, varying from 0.1 pm/V to the maximum 27.3 pm/V (d_{16} and d_{35}). The high predicted shear value and doubling of the longitudinal response in the Hyp-Phe-Phe crystal highlights that simple chemical modifications can induce piezoelectric response in biomaterials that exceeds that of many inorganic crystals (Fig. 2e). The maximum calculated response for Hyp-Phe-Phe is as large as the d_{31} value of poled PVDF....”

Page 9, lines 21-23:

“For comparison (Fig. 2f), KNN-based ceramics have exhibited voltage constants of $g_{33} = 40$ mV m/N. Other studies have reported values of up to 540 mV m/N (g_{33}) for single crystals of BiB_3O_6 .”

Page 10, lines 5-8:

“The benefit of predicting the full piezoelectric tensor with DFT and screening for individually high tensor components is that we can design devices and orientate biomolecular crystals and assemblies in a way that maximizes output.”

Materials added to Supplementary Table 1:

Biomaterials: Silk

Inorganic Materials: $(\text{Na}_{0.5}\text{Bi}_{0.5})\text{TiO}_3$ - BaTiO_3 - $(\text{K}_{0.5}\text{Na}_{0.5})\text{NbO}_3$ single crystals, $(\text{K}_{0.88}\text{Na}_{0.12})\text{NbO}_3$ films

2D/nanostructured materials: MnO_2 nanorods/PVDF hybrid films, PMMA/ZnO nanowires, ZnO nanorods, PZT nanoshells, GaN nanowires, SnSe, GeS

Class of material	Material	Piezoelectric coefficient (pm/V)	Supplementary Reference
Biomaterials	Pro-Phe-Phe (DFT prediction)	1.9 (d ₂₂)	This work
	Hyp-Phe-Phe (DFT prediction)	27.3 (d ₃₅)	
	Hyp-Leu-Phe (DFT prediction)	3.6 (d ₃₄)	
	Pro-Phe-Phe (QFN-AFM measurement)	2.2 (d ₃₃)	
	Hyp-Phe-Phe (QFN-AFM measurement)	4.0 (d ₃₃), 16.1 (d ₃₄)	
	Bone	0.1 (d ₃₃)	5
	Collagen microfibril	2.64 (d ₃₃)	6
	Type I collagen	1.1 (d ₁₅)	7
	Fish skin collagen	5.6 (d ₃₃)	8
	Collagen triple helix	12 (d ₁₄)	9
	Collagen peptide	10 (d ₁₄)	10
	M13 bacteriophage film	11 (d ₃₃)	11
	Vertically aligned phage nanopillars	6 (d ₃₃)	12
	α -helical poly(α -amino acid)	25 (d ₃₃)	13
	Poly-L-lactic acid	15 10 (d ₁₄)	14
	FF	9.9(d ₃₃)	15
	FF aligned	18(d ₃₃)	16
Silk thin film	56.2(d ₃₃)	17	
Polymers	Bi-axial Poled polyvinylidene-fluoride (PVDF)	32.5(d ₃₃)	18
Inorganic Materials	CdS	14 (d ₁₅)	19
	ZnO	12.4 (d ₃₃)	20
	LiNbO ₃	69 (d ₁₅)	21
	AlN Thin Films	5.1 (d ₃₃)	22
	(Na _{0.5} Bi _{0.5})TiO ₃ -BaTiO ₃ -(K _{0.5} Na _{0.5})NbO ₃	840 pC/N (d ₃₃)	23

	(K _{0.88} Na _{0.12})NbO ₃ films	71 pC/N (d ₃₁)	24
2D / nano- structured materials	MnO ₂ nanorods/PVDF hybrid films	38 pC/N (d ₃₃)	25
	PMMA/ZnO NWs	26 pC/N (d ₃₃)	26
	ZnO Nanorods	9.5(d ₃₃)	27
	PZT Nanoshells	90(d ₃₃)	28
	GaN Nanowires	12.8(d ₃₃)	29
	SnSe	251 (d ₁₁)	30
	GeS	75 (d ₁₁)	31

2. Reviewer's comment:

Second, the computational works provide sufficient explanations and supplement the studies. Therefore, the modeling of the molecular structure of the material and the simulating conditions of the surroundings are crucial. What is the pH condition? The authors should provide more information about the experimental methods because the dipole or polarization of the material will be easily affected by the pH conditions. Also, how did the authors defined the surrounding conditions regarding the water contents? The water contents (humidity) may strongly affect the piezoelectric property, molecular structure, and further mechanical strength of the materials. Moreover, how was the molecular structure of the peptides optimized? This is because the simulation works may show different results depending on the optimized structure of the materials. Therefore, more detailed explanations of how the simulations were performed will help the reader to better understand the methods and improve the usefulness of the work.

Authors' response:

We thank the reviewer for their expert comments on the molecular modelling, and we agree that the summary we supplied in the main body of the article may lead to confusion for readers, as we employ various modelling techniques so that we can calculate the piezoelectric coefficients using

DFT and the dynamics of the supramolecular packing using classical MD. To address this issue, we have added further descriptions and details of the modelling we are performing, both in **Results and Discussion** and in **Methods** sections. We have also discussed the techniques in the introduction. Both in the crystal unit cells used for DFT and the supramolecular assemblies used for classical MD, the pH is set by the physiological protonation state of the amino acids to match the pH of 7.4 used for the crystallization of the peptide.

Bulk water is only considered for the MD simulations, in which a large water box surrounds a supercell generated from the peptide unit cell. We used this model to monitor the room temperature dynamics of the supramolecular packing in the peptide assembly, starting from the X-ray crystal structure packing. No piezoelectric calculations were performed on hydrated crystals as, so far, no experimental structures are available for such systems.

In both DFT and MD, the peptide structure is optimized by energy minimization of the starting X-ray structure to the most thermodynamically stable state, and for MD, the time evolution of the structure is calculated using the CHARMM36m classical force field in the constant pressure-temperature ensemble as described (now in more detail) in Methods. These MD simulations will also make it possible to observe the potential crumbling of the peptide crystals in water, in future work which will allow us to characterize the balance between peptide-peptide and peptide-water interactions in tuning the mechanical properties for potential applications beyond solid-state power generators in, *e.g.*, implantable devices for sustained drug delivery *in vivo*.

Changes made to the manuscript:

Page 3, lines 21-23; Page 4, lines 1-7

Molecular modelling, in particular Density Functional Theory (DFT), has emerged as an effective tool for predicting and rationalizing the piezoelectric response of materials, from classic inorganic crystals^{27,28} to polymers²⁹ and novel 2D structures^{30,31}. Our previous work has utilized DFT to accurately predict the elastic and piezoelectric tensors of amino acid^{10,11}, peptide²⁵, and biomineral crystals³². DFT can be used to complement techniques such as Piezoresponse Force Microscopy (PFM)^{33,34}, or used in stand-alone predictive modelling studies of nanoscale electromechanical phenomena³⁵. Classical molecular dynamics simulations are also widely used to study the kinetics

of piezoelectric systems³⁶, and are particularly effective at studying systems in a liquid environment³⁷, and over a range of temperatures³⁸.

Page 8, lines 5-9

Having verified helical conformation, lateral periodicity and high mechanical robustness similar to collagen, we used DFT to predict the elastic, dielectric and piezoelectric constants of the tripeptides (Fig. 2 & Table 1). Calculations were carried out on the obtained solid-state XRD crystal structures, which contained no waters of crystallization. Full details of the computational methodology can be found in the **Methods** section.

Page 14, lines 2-5

These simulations are performed on a nanocrystal of each peptide immersed in a large water box to model bulk solvation, as described in the **Methods** section. Molecular dynamics are calculated at constant room temperature and atmospheric pressure at physiological pH of 7.4.

Page 25, lines 1-16

Molecular dynamics (MD) simulations were performed with the Gromacs 2018.4 package. All bonds involving a hydrogen atom were constrained using the LINCS⁷² algorithm which allowed use of an integration timestep of 2 fs in the leapfrog integrator,⁷³ and coordinates were saved every 10 ps. Long-range electrostatics were treated by the Particle Mesh Ewald (PME) method.⁷⁴

All systems were minimized for 100,000 steps, and heated progressively from 0 to 300K for a total 3 ns at constant volume, followed by 100 ns of equilibration at constant pressure and temperature. During equilibration the reference temperature was set at 300K with a time constant of 1ps and the reference pressure was set at 1 bar with a time constant of 5 ps, using the Berendsen⁷⁵ thermostat and barostat, respectively. The subsequent production phase of dynamics was carried out in the constant pressure and temperature NPT ensemble, with the reference pressure set at 1 bar with a time constant of 5 ps using the highly accurate Parrinello-Rahman barostat,⁷⁶ and all molecules were coupled separately in groups to an external heat bath set at 300K with a coupling time constant of 1 ps using the velocity-rescale thermostat.⁷⁷ The pH is set by the protonation state of the amino acids at the physiological pH of 7.4 used in the crystallization experiments.

The unit cell space group and translation symmetries are conserved in the DFT calculations used to compute the electromechanical properties of the single crystals. They are not conserved in the MD simulations, as the crystal supercell or block is immersed in a large box of pre-equilibrated bulk water. These MD simulations were used to monitor the evolution of the supramolecular π - π and H-bond contacts in the tripeptide assemblies in the central region of the block, to obtain time average values and error bars for the contacts identified in the starting X-ray structure.

3. Reviewer's comment:

Third, from the last sentence of page 5 to the first sentence of page 6, what is the solid evidence in that the neighboring helical structural modules connected in a parallel orientation with the interface of the dimer were stabilized by the aromatic zipper structure built from pi-pi interactions between the Phe side chains. Since this is crucial to explain the extended helical-like molecular organization of the material, simply providing an assumption by referring other studies to explain the authors' theory seems less convincing. Therefore, the authors should provide solid evidence for the explanations.

Authors' response:

We thank the reviewer for urging us to more deeply analyze and fully exploit the molecular dynamics simulations, which has increased the usefulness of the present work. The analyses of the MD trajectories we presented in the original submitted manuscript characterized the stability of the stacking layers of tripeptides in the supramolecular assembly. We demonstrated that an ordered and robust structure is created by π - π stacking between phenyl rings. However, we did not show similar results for the extended zipper motif. In the revised manuscript, we characterize the stability of the zipper for both Pro-Phe-Phe and Hyp-Phe-Phe. Supplementary Fig. 17 in the revised Supporting Information shows the zipper structure created by the large-area intermolecular contacts provided by the high-density of Phe residues in the tripeptide assembly. The tightly-bound zipper persists during the entire 200 nanoseconds of room temperature dynamics for both tripeptides and the structures shown are from the final frame of the trajectories. The strong computed interaction energies during 200 ns of dynamics (Supplementary Fig. 18) confirm the

stability of zipper. Finally, we performed the same structural analysis for the zipper as was presented for the interlayer stacking. Supplementary Fig. 19 in the SI shows the distribution of the contact distances between pairs of phenyl rings on neighboring molecules that hold together the zipper motif, and their corresponding orientations. For both tripeptide assemblies the distributions are narrow and centered close to the contact distance in the X-ray crystal structures, which confirms the stability and durability of the zipper as a structure-building motif. Supplementary Fig. 19 shows density maps of the distance-angle landscape, illustrating the predominant major population of pi-stacked contacts in the zippers.

Changes made to the manuscript:

Page 14, line 18-21

The evidence for the stability of the zipper for both Pro-Phe-Phe and Hyp-Phe-Phe have been obtained from both MD simulations and crystal structures, which confirm the rigidity and durability of the zipper as a structure-building motif (Supplementary section 4, Supplementary Fig. 17-20).

Changes made to the Supplementary Information:

Supplementary Section 4, Pages 23-25

Supplementary Fig. 17 shows the final computed structure of the stable, persistent, and tightly-knit zipper motif in (a) Pro-Phe-Phe and (b) Hyp-Phe-Phe following 200 ns of molecular dynamics.

Supplementary Fig. 17 | Stick representation of the zipper motif that stabilizes the tripeptide assemblies. The phenyl rings involved in the zipper are depicted in gray for the second residue and orange for the third residue of the tripeptide in (a) Pro-Phe-Phe and (b) Hyp-Phe-Phe. The green shapes indicate the contacts that are selected in the monitoring of zipper stabilization in Supplementary Fig. 18.

The stability of the zippers is confirmed by the net favorable computed interaction energies shown in Supplementary Fig. 18. The extended pi-pi zipper seam creates a strong and stable intra-layer structure, which couples with the stable interlayer π - π stacking (Fig. 3) to create the robust, mechanically-stable tripeptide assemblies.

Supplementary Fig. 18 Interaction energies computed over 200 ns within the zipper of Pro-Phe-Phe (blue) and Hyp-Phe-Phe (red), using the 6-Phe sampling units marked in Figure Supplementary Fig. 17. The orange and green curves are the running averages for Pro-Phe-Phe and Hyp-Phe-Phe, respectively.

Supplementary Fig. 19 shows the corresponding distribution of the distance between pairs of phenyl rings across the zipper motif and their relative orientation during the final 50 ns of dynamics. For both tripeptide assemblies the distance distributions are narrow and centered close to the contact distance in the X-ray crystal structures, which together with the ordered orientations confirms the strength and stability of the zipper. Supplementary Fig. 19 c,d shows density maps of the distance-angle landscape, illustrating the predominant major population of ordered pi-stacked contacts in the zippers.

Supplementary Fig. 19 | Distributions during the final 50 ns of dynamics of the distance (a) and angle (b) between pairs of phenyl rings that make the zipper. Data for Pro-Phe-Phe is plotted in blue and data for Hyp-Phe-Phe is plotted in red. The distance-angle distributions are characterized in more detail in the density maps in panels (c) Pro-Phe-Phe and (d) Hyp-Phe-Phe.

Additional authors' response:

In addition to these new modelling results, we have also now mapped the unit cell of a single crystal on to the macroscopic crystal morphology. This confirms that the *a*-axis corresponds to the long axis of the crystal and the *b*-axis represents its width. As can be seen from the crystal packing (Fig. 1f), the helical strands grow along the *b*-direction and adjacent helices are stacked laterally with respect to each other in the *c*-direction through stabilization of the interface by the aromatic zipper structure. In the perpendicular plane (*a*-direction), neighboring helices interacted in a

parallel pattern through intermolecular hydrogen bonds, stacking into a closely packed helical sheet to produce the elongated structure. Therefore, in the crystal packing of Hyp-Phe-Phe, helical strands run perpendicular to the long axis and are stabilized by the aromatic zipper structure, whereas helical sheets run parallel to the long axis and H-bonds provide the stabilization. This analysis further confirms the organization of helices and identifies the important supramolecular interactions that direct and stabilize the observed tightly-packed assemblies.

Changes made to the Supplementary Information:

Supplementary Section 4, Page 25

“The measured unit cell parameters of a single crystal with respect to the crystal morphology also helps confirm the specific organization of helices that stabilize the crystal (Supplementary Fig. 20).”

Supplementary Fig. 20 | Unit cell measurement of the crystal with respect to crystal morphology. Single crystal is shown mounted on a MiTeGen loop. The crystal is highlighted by the white box and the respective cell axes are marked in red. The morphological long axis of the crystal is aligned along the crystallographic *a* axis of the unit cell. Size of the crystal is 0.262 x 0.090 x 0.016 mm.

4. Reviewer's comment:

Fourth, for the PFM measurements, the authors are only providing the piezoelectric coefficients of different peptide assemblies without any experimental images including topography, amplitude, deflection, and phase. These detailed experimental results should be supplemented to rule out the artifact-induced piezoelectric property of the materials.

Authors' response:

We thank the reviewer for raising this point. Unfortunately, due to the small size of the peptide single crystals, conventional PFM imaging of topography, amplitude, phase, *etc.* was not possible, as the crystals moved during imaging attempts. In line with best practice, we carried out PFM point measurements where the probe was brought into contact with the single crystal and held stationary while the applied voltage was varied and the piezoresponse recorded. The images have been provided in Supplementary Fig. 11 and 12. These were used to generate plots similar to Figure 2h, which were then used to create statistical distributions of the response. All the plots are included in Supplementary Fig. 6-13. The measurements were carried out with the probe held stationary relative to the sample and at a low frequency (21 kHz), which ensured that any artefacts resulting from topographic crosstalk or resonance enhancement of the signal would be negligible. Additionally, we provide both positive and negative controls in the SI to further justify the method. We have added extra detail to the revised manuscript to reflect these points.

Changes made to the manuscript:

Page 12 line 22-23; page 13, lines 1-14

Due to the small size of the peptide single crystals, conventional PFM imaging of topography, amplitude, phase, *etc.*, was not possible. The crystals moved during imaging attempts. In line with best practice techniques^{34,53}, we then carried out PFM point measurements where the probe was brought into contact with the single crystal and held stationary while the applied voltage was varied and the piezoresponse recorded. These were used to generate plots similar to Figure 2h, which were then used to create statistical distributions of the response. The measurements were carried out with the probe stationary relative to the sample and at a low frequency (21 kHz), which minimized artefacts resulting from topographic crosstalk or resonance enhancement of the signal. Stiff probes with a spring constant of 5-6 N/m were used to mitigate electrostatic and flexoelectric contributions³⁴. All measurements were carried out at 20 °C and 40% RH ambient laboratory conditions to ensure uniformity in the measurement conditions. Identical point measurements were carried out on both positive and negative controls to verify the accuracy of the technique and to rule out any instrumental backgrounds or parasitic effects contributing to the signal, as described in detail in Supporting Information section 3.

5. Reviewer's comment:

Fifth, for the characterization of the peptide-based power generator, the authors should provide more detailed information about the device and the characterization results. In Figure 4a, the schematic image shows that the peptide assemblies are deposited on Kapton, and PDMS is also included between the electrodes. How can the authors be sure that the peptide assemblies combined with Kapton and PDMS show the pure piezoelectric property of the power generator? These materials may affect not only the electrical characterization, but also to the effective force application to the device. Therefore, the applied force indicated in Figures 4h and 4i may not be the effective force that was applied to the device. Moreover, the authors should also provide the voltage and current based on the different resistances to see the electrical characteristics of the power device. Furthermore, since the authors are also emphasizing the high mechanical property of the material, the durability test of the power generator should be supplemented.

Authors' response:

We thank the reviewer for raising these points and providing us with the opportunity to clearly represent the stacking arrangement of the devices and their efficiency. We have now included a detailed description of the device fabrication in the modified **Methods** section, which clarifies the device building process.

Changes made to the manuscript:

Page 27, line 3-17

“Fabrication of peptide-based power generator. Two $0.7 \times 0.7 \text{ cm}^2$ Ag layers were deposited on two $1.2 \times 1.2 \text{ cm}^2$ silicon substrates that served as the top electrode and the bottom electrode. The smooth surface of a Kapton tape was first coated with a PDMS layer of thickness 0.40 mm. A strip of double-sided tape was then attached to the top surface of PDMS. A $0.7 \times 0.7 \text{ cm}^2$ hole was cut through the Kapton/PDMS/double-sided tape structure. The Kapton tape was attached to one silicon substrate with the Ag electrode exposed to the hole. The hole was then filled densely with the Phe-Phe, Pro-Phe-Phe and Hyp-Phe-Phe peptide assemblies, respectively. The other silicon substrate sputtered with silver film was then attached to the double-sided tape to complete the device fabrication without leaving any gap in the nanogenerator. In order to prevent cracking of the substrate due to the impact force from the linear motor, the top substrate was drop-coated with

a PDMS damping layer. Two copper wires were connected to the silver electrodes using carbon ink (JELCON CH-8 MOD2) to complete the device fabrication.”

We have now included an additional figure in supplementary information, which illustrates the different layers of the device more clearly.

Supplementary Fig. 21 | **a**, Schematic of the proof-of-concept power generator with peptide assemblies as the active components. **b**, Schematic cross-section diagram of the device architecture.

From the detailed description of device fabrication and schematic of the layered structure, it can be clearly observed that Kapton and PDMS were not included between the two Ag electrodes. Only peptide assemblies were put between the two Ag electrodes. Therefore, the obtained output from the device comes solely from the piezoelectric response of the peptide and not from the structural supports in the power generator.

We have also modified the schematic configuration of energy generation measurement set-up in the main manuscript (Fig. 4a) to more accurately represent the stacking layers.

According to the reviewer’s suggestion, we have performed the durability test of the two different nanogenerators under applied cyclic forces of 17 N with periodic displacement of 1,000 cycles for over 60 min. Our data demonstrates the stable performance of the device and indicates high durability of the peptide-based power generator.

Changes made to the manuscript:

Page 18, lines 11-14

“Furthermore, the high mechanical rigidity suggests that power generation can be sustained under a cyclic force (17 N) (Supplementary Fig. 23) and the output voltage showed no degradation over 1000 press/release cycles for more than 60 minutes, indicating the high durability of the peptide-based devices.”

Supplementary Fig. 23| Stability testing of Hyp-Phe-Phe peptide nanogenerator. The open-circuit voltages from two different nanogenerators (a-c) and (d-f) were measured under applied force 17 N upon periodic displacement for over 60 min. The full datasets are plotted in the left hand panels with detailed views of periodic voltage response for portions (highlighted in black boxes) shown in the middle and then right panels. The peak to peak magnitude of the open-circuit voltage remained constant, demonstrating the stable performance of the device.

According to the reviewer’s suggestion, we have also studied the output performance of the nanogenerator under different applied load resistances and the results are now included in the revised manuscript.

Changes made to the manuscript:

Page 18, lines 14-19

“Finally, the voltage output characteristic was measured with different external load resistors connected to a nanogenerator while it was repeatedly deformed, and the result is shown in Supplementary Fig. 24. The output voltage continually rises with the growth of load resistance, demonstrating the electrical characteristics of the power device and illustrating its potential for practical applications.”

Supplementary Fig. 24| Characterization of Hyp-Phe-Phe based nanogenerator. The open-circuit voltage measured upon applying increasing external load resistances 1 MΩ (a), 5 MΩ (b), 10 MΩ (c), 50 MΩ (d), 100 MΩ (e) and 500 MΩ (f) at the constant applied force of 17 N. g) The average voltage output as a function of external load resistance.

6. Reviewer's comment:

Sixth, the use of expressions such as “quite, slight, slightly, very small, similar, significantly, only, only very, etc.” leads to an unclear and ambiguous interpretation of the piezoelectricity of the materials. When making comparisons, the authors should clearly deliver the meanings by presenting numerical values or only large-small comparisons.

Authors' response:

We agree with the referee, and apologize for the ambiguity. We have removed or clarified all such instances in the revised text.

Changes made to the manuscript:

Page 8, line 1: “quite a similar” has been changed to “similar”

Page 8, line 11: “slight increase” - the values are now mentioned (3.3 vs. 3.1)

Page 10, line 17: “slightly larger” - the values are now provided (2.3 eV vs. 2.0 eV)

Page 12, line 6: “Only very small” - the numerical values have been included (24 GPa vs. 28 GPa)

Page 12, line 9: “slightly below” - the values have been mentioned in the line (12 GPa vs. 14 GPa)

Pg. 16 line 18: “which is significantly higher than reported peptide and inorganic alternatives (Supplementary Table 9)” – the SI Table 9 contains all the numerical values of peptide and inorganic materials.

Page 16, line 20: “which is significantly higher than the output current obtained from nanogenerators based on M13 bacteriophage virus (6 nA) or fish skin collagen (1.5–20 nA) (Supplementary Table 9)” - along with the values given in parentheses, all other numerical numbers have been provided in the SI Table 9.

Page 16, line 23: “similar short circuit current” - the value has been included in the revised text.

Reviewer 2

Reviewer's opening remarks:

This manuscript reports two Phe-Phe-derived peptide crystals that demonstrate high piezoelectricity. Using PFM, together with DFT and MD simulations, the authors showed that simple sidechain engineering of tripeptides, such as adding a hydroxyl group, could dramatically increase peptide crystals' piezoelectric response. The authors also demonstrated a large scale tripeptide-based generator that can produce current and voltage. While reading the manuscript, I was impressed by these tripeptides' extremely high piezoelectric constants. For example, the piezoelectric voltage constant of Hyp-Phe-Phe is even higher than that of widely used PZT. However, I have some concerns related to the origin of their measured electromechanical response using PFM and the large scale peptide generator, since other mechanisms could also lead to piezoelectric-like behaviors. Here, I provide comments that could be used to verify tripeptides' piezoelectric properties and to further improve the manuscript.

Authors' response:

We are happy that the referee was impressed by the high piezoelectric response we engineered in our tripeptide materials. We are grateful for their expert review and below we answer all comments point-by-point including details of all changes and additions we made to our revised manuscript.

1. Reviewer's comment:

There have been many discussions in the field that PFM may not be a reliable tool to quantify nanomaterials' piezoelectricity (Gruverman, Nature Communications 10, 1661 (2019)), because many non-piezoelectric materials, such as HfO₂ (Kim, ACS Nano 6, 7026 (2012); Blake, ACS Nano 9, 6484 (2015)), can mimic piezoelectric behaviors in PFM measurements, and it is challenge to characterize the true piezoelectric constants due to local electrostatic effects and non-uniform electric fields. Also, the measurement could be very sensitive to local environmental conditions, i.e. temperatures and humidity. Clearly, the only voltage vs. displacement PFM results presented in the paper is not sufficient to proof peptides' piezoelectricity. The authors should perform in-depth experiments and data interpretations to verify and quantify the piezoelectric effects. For

example, as suggested in Blake, ACS Nano 9, 6484 (2015), showing hysteresis loops of PFM amplitudes and phases with various electric fields could dramatically improve the accuracy of measured piezoelectricity. In addition, the authors should provide more information: If the measured sample is a single crystal or not? If not, how to compare the PFM data to simulation results? Do these peptides need any polarization to exhibit piezoelectricity, and why?

Authors' response:

We agree with the reviewer that there are challenges associated with quantification in PFM. We have recently published a tutorial style article that addresses many of these concerns, in particular, the application of the technique to biomolecular crystals (J. O'Donnell *et al.* 2020, Applied Materials Today, 21,100818), and in the current study we follow the PFM methodology outlined therein. Regarding the works referenced by the reviewer, the vast majority of the concerns raised in these papers relates to the difficulty of measuring ferroelectricity using PFM and the resulting hysteretic behavior that can occur due to non-ferroelectric mechanisms. We share these concerns regarding the difficulties of using PFM to detect and measure ferroelectricity, and we have not attempted to carry out any ferroelectric measurements in this work. Nor do we make any claims regarding the ferroelectric/non-ferroelectric nature of the materials in this work. We have simply used PFM to measure the voltage-induced deformation of the material and have found it to be linear, suggesting piezoelectricity. As mentioned in Gruverman, Nature Communications 10, 1661 (2019), we agree that other effects such as electrostatic or flexoelectric interactions can mimic a true piezoelectric response and so influence measurements of this type. We used stiff probes with spring constants of 5-6 N/m along with a large contact force to mitigate these effects and ensure that the observed linearity is a true indicator of piezoelectricity in the material response, as described in detail in the recent Applied Materials Today methodology paper. We further ensure the accuracy of the technique in the present study by using both positive and negative controls to show that any residual background effects are not playing a large role. Finally, all measurements were carried out at standard ambient laboratory conditions of 20 °C and 40% relative humidity to ensure uniformity across all measurements.

With regards to the suggestion that “showing hysteresis loops of PFM amplitudes and phases with various electric fields could dramatically improve the accuracy of measured piezoelectricity”, we have opted to avoid the use of DC fields and hysteresis loop acquisition for the reasons outlined

above (and discussed more in the publications the reviewer has cited). The stationary PFM point measurements we have used here simply monitor the sample deformation under a varying applied voltage, which provides a much simpler and unambiguous measurement of piezoelectricity. The measured samples are single crystals and do not require any additional polarization as they crystallize in non-centrosymmetric space groups and so are inherently piezoelectric without requiring polling. We have added extra detail to the PFM section and cited the latest work on application of the technique to biomolecular materials to further reinforce the need for the precautions we take and the resulting utility of the technique when these precautions are taken. We agree with the reviewer and the cited studies that ferroelectric measurements are not reliable with PFM, which is why we have not attempted to carry out any hysteresis measurements involving varying DC fields.

Changes made to the manuscript:

Page 13, lines 8-14

“Stiff probes with a spring constant of 5-6 N/m were used to mitigate electrostatic and flexoelectric contributions³⁴. All measurements were carried out at 20 °C and 40% RH ambient laboratory conditions to ensure uniformity in the measurement conditions. Identical point measurements were carried out on both positive and negative controls to verify the accuracy of the technique and to rule out any instrumental backgrounds or parasitic effects contributing to the signal, as described in detail in Supporting Information section 3.”

2. Reviewer's comment:

For the peptide-based nanogenerator, it might not be appropriate to claim that the measured voltage and current outputs originate from the piezoelectric effect, and switching the connect is not sufficient to exclude other electromechanical effects. For example, triboelectric generators with similar sandwiched electrode/PDMS/electrode or electrode/PDMS composite/electrode structures can produce equivalent or much higher voltages and currents with similar patterns, such as that demonstrated in Chen. ACS Applied Materials & Interfaces 8, 736 (2016) and He, Nano Research 9, 3714 (2016). Given that tripeptides could exhibit extremely high voltage constants, the authors should have a rational explanation about why the measured voltage output from the nanogenerator

is relatively low. In addition, measuring the power output with various external loads will better illustrate its potential for practical applications.

Authors' response:

We thank the reviewer for raising this point. We fully agree with the reviewer that the triboelectric effect can play an important role in the study of nanogenerators, and we have been very careful to eliminate the triboelectric effect. We have included more detailed description of the device fabrication and additional images of the layered structure of the nanogenerator to clarify the stacking pattern of the device and the characteristics of the output.

Page 27, line 3-17

“Fabrication of peptide-based power generator. Two $0.7 \times 0.7 \text{ cm}^2$ Ag layers were deposited on two $1.2 \times 1.2 \text{ cm}^2$ silicon substrates that served as the top electrode and the bottom electrode. The smooth surface of a Kapton tape was first coated with a PDMS layer of thickness 0.40 mm. A strip of double-sided tape was then attached to the top surface of PDMS. A $0.7 \times 0.7 \text{ cm}^2$ hole was cut through the Kapton/PDMS/double-sided tape structure. The Kapton tape was attached to one silicon substrate with the Ag electrode exposed to the hole. The hole was then filled densely with the Phe-Phe, Pro-Phe-Phe and Hyp-Phe-Phe peptide assemblies, respectively. The other silicon substrate sputtered with silver film was then attached to the double-sided tape to complete the device fabrication without leaving any gap in the nanogenerator. In order to prevent cracking of the substrate due to the impact force from the linear motor, the top substrate was drop-coated with a PDMS damping layer. Two copper wires were connected to the silver electrodes using carbon ink (JELCON CH-8 MOD2) to complete the device fabrication.”

We have now included an additional figure in revised supplementary information, which illustrates the different layers of the device more clearly.

Supplementary Fig. 21 | **a**, Schematic of the proof-of-concept power generator with peptide assemblies as the active components. **b**, Schematic cross-section diagram of the device architecture.

As shown in Supplementary Fig. 21, the top Ag electrode and the bottom Ag electrode are in contact with only the piezoelectric peptide assemblies for the energy generation. The PDMS does not sit between the Ag electrodes, which eliminates any triboelectric effect from the PDMS layer.

We have also modified the schematic configuration of energy generation measurement set-up in the main manuscript (Fig. 4a) to more accurately represent the stacking layers.

As the peptide assemblies are only included between the two electrode, thus the measured voltage and current outputs originate from the true piezoelectric effect of the peptides. Therefore, the principle of the fabricated devices here and their outputs are different from the triboelectric based nanogenerators reported by Chen, *ACS Applied Materials & Interfaces* 8, 736 (2016) and He, *Nano Research* 9, 3714 (2016). We have mentioned this in the Methods section and cited the referred articles in the revised manuscript (See references 81,82). We believe the relatively low voltage output from the nanogenerator is mainly due to the random orientation of the Pro-Phe-Phe, Phe-Phe and Hyp-Phe-Phe peptide assemblies in the nanogenerator. The nanogenerator could potentially generate much higher voltage output if all Pro-Phe-Phe, Phe-Phe and Hyp-Phe-Phe peptide structures could be aligned in one direction, and this has been mentioned in the manuscript (page 18, line 20-22), we are beginning to work toward engineering solutions which will take time and which are beyond the scope of the present work.

According to the reviewer’s suggestion, we have now studied the output performance of the nanogenerator under different applied load resistances and the results are now included in the revised manuscript showing the promise for future applications.

Changes made to the manuscript:

Page 18, lines 14-19

“Finally, the voltage output characteristic was measured with different external load resistors connected to a nanogenerator while it was repeatedly deformed, and the result is shown in Supplementary Fig. 24. The output voltage continually rises with the growth of load resistance, demonstrating the electrical characteristics of the power device and illustrating its potential for practical applications.”

Supplementary Fig. 24| Characterization of Hyp-Phe-Phe based nanogenerator. a-f, The open-circuit voltage measured upon applying increasing external load resistances 1 M Ω (a), 5 M Ω (b), 10 M Ω (c), 50 M Ω (d), 100 M Ω (e) and 500 M Ω (f) at the constant applied force of 17 N. g) The average voltage output as a function of external load resistance.

3. Reviewer's comment:

Page 6-7: The QNM results could be largely affected by peptide fibres' surface geometry, and it is very sensitive to topography changes. For example, a smaller contact area between the AFM tip and the sample will lead to a higher measured Young's modulus. The authors should be more careful about the claims in Fig 1g-I, and tip/surface contacts should be analyzed.

Authors' response:

We thank the reviewer for the comments. We believe that by using carefully calibrated tips with a radius almost two orders of magnitude smaller than the fibril, we employ a technique that is able to record accurate and reproducible mechanical data for these fibrils.

The tip radius of the AFM probes used is approximately 8 nm, and the fibrils are approximately 500 nm in diameter, therefore the contact between tip and fibril should be consistent over multiple positions and scans. We (and other authors) have performed similar QNM-AFM experiments on much smaller fibrils with success and repeatability (see Adamcik *et al.*, 2012 *Nanoscale*, 4, 4426, Dharmadana *et al.*, 2018, *Nanoscale*, 10-18195 and others for examples). We carefully calibrated each AFM cantilever to quantify its deflection sensitivity, spring constant, tip radius and maximum indentation depth. This procedure allowed us to reproducibly measure Young's modulus values in the range of 60-90 GPa (the large variance is due to the observed lateral periodicity) using multiple different tips and different samples. A detailed description of the calibration protocol used is provided in the supplementary information (near Supplementary Fig. 3). We have added a few sentences to this protocol to specify the typical tip radius and also to emphasize that in a typical experiment, the maximum indentation depth is less than 1% of the fibril thickness which eliminates any substrate effect on the measured Young's modulus.

Regarding being more careful about the claims of our QNM-AFM results, we have rewritten the sentence “The measured Young’s modulus of the fibrils was approximately 85 GPa with quite large variation.....” to be more exact by stating “The measured Young’s modulus of the fibrils varied in the range of 60-90 GPa, producing the same order of magnitude mechanical stiffness as recorded by AFM nanoindentation in the corresponding single crystal” (Page 7, line 11-13)

4. Reviewer’s comment:

Page 6: Fig. 1g is confusing. The substrate image is identical to its topography image in Fig. 1d, but the legend only shows Young’s modulus.

Authors’ response:

We apologize for this admittedly confusing AFM image, and thank the referee for alerting us to this. The original panel 1g was a composite image of the topographical background from figure 1d and the DMT channel from figure 1i. On reflection we see no benefit in presenting the data in this way and agree it is confusing. Therefore, we have removed this image and replaced it with two separate images: we now show topography in Fig. 1d and a new version of Fig. 1g shows only the DMT channel generated by the QNM-AFM imaging mode:

5. Reviewer’s comment:

Page 6: Add a legend in Fig. 1i.

Authors’ response:

In order to simplify the presentation of data, Fig. 1i has been removed.

6. Reviewer's comment:

Page 10: The authors could uniform the units of the calculated d_{ik} constants in Fig. 2a-b and the measured ones in Fig. i-j.

Authors' response:

In order to avoid confusion, we now use pm/V throughout, and no longer use the equivalent unit pC/N.

7. Reviewer's comment:

Page 10: Fig. 2h can be changed to electric field vs. strain.

Authors' response:

The units of piezoresponse and applied voltage in Fig. 2h are nanoamperes and volts, respectively. This reflects the quantities measured by PFM in the experiment. These quantities are then used to convert the slope of this line to a measurement of the piezoelectric coefficients (please see Supplementary Information). We acknowledge the reviewer's observation that converse piezoelectricity is a measurement of strain as a function of electric field, but in the context of the PFM experiment and the data collected, it is not practical to display these units on the graph.

8. Reviewer's comment:

SI section: Fig. S1bc need to show their substrates.

Authors' response:

We thank the reviewer for this suggestion. Accordingly, we have updated Fig. S1 with new images that now show the substrates.

Supplementary Fig. 1 | **a**, AFM images of Hyp-Phe-Phe on silica substrate showing the presence of fibre structure along with some nanocrystals. The vertical width of the image is 15 μm . **b**, Topographic AFM images of Hyp-Phe-Phe fibrils, Z-scale = 133 nm (left side) and Z-scale = 250 GPa (right side).

Response to Editorial comments:

1. We have completed the Editorial policy checklist form and uploaded with the resubmission.
2. For the new peptide molecules, evidence of sample purity based on HPLC analysis and evidence of identity based on NMR and Mass spectral characterization has been included with the revised manuscript.
3. New structures of peptide from crystallographic analysis has been accompanied by a standard crystallographic information file (cif). The structure factors and structural output has been checked using IUCr's CheckCIF routine and a PDF copy of the output has been included with the resubmission. Crystallographic data has been submitted to the Cambridge Structural Database and the deposition number is referenced in the manuscript.
4. The manuscript is accompanied with a "Data Availability" section after the Methods section but before the References with all the details about the accessibility of data.
5. A file containing the raw data in a single Excel file has been provided with the resubmission and this has been mentioned in the "Data Availability" section.

REVIEWER COMMENTS

Reviewer #1 (Remarks to the Author):

The revised manuscript addressed well for the critiques that this reviewer raised. Therefore, current manuscript can be publishable for Nature Comm. n/a

Reviewer #2 (Remarks to the Author):

I appreciate the authors' efforts to address my concerns and comments. I totally understand the difficulty in probing nanoscale materials' piezoelectric properties. However, since the piezoelectricity of peptide crystals is a major claim of this paper, I do think that it is necessary to clarify the origin of these measured piezoelectric properties, especially those measured by PFM. Thus, I cannot recommend publication before addressing the following issues.

To be clear, my previous comments suggest additional PFM experiments and data interpretations to verify and quantify the piezoelectric effects, rather than ask the authors to proof materials' ferroelectric nature. As mentioned by the authors, the suggested articles, including Gruverman, Nature Communications 10, 1661 (2019), do give reasons why current PFM methods are not reliable for piezoelectricity measurements, including the voltage-induced deformation. Thus, using a probe with a certain stiffness at a fixed frequency will not eliminate other effects. For the same reason, together with the difference in surface charge, sizes, and shapes, the positive and negative controls cannot ensure the accuracy of the measurements.

I understand the difficulty in measuring nanoscale crystals that are movable on substrates, but the PFM experiment, together with the very low g_{33} from the nanogenerator, raise a lot of concerns for me. For example, the authors applied up to 20 V across a ~ 180 nm (Fig.1d) thick peptide crystal in the ambient environment. Based on my quick calculation, the resulting electric field reaches 111 kV/mm, which is extremely high comparing to breakdown electric fields of ~ 3 kV/mm for dry air and ~ 60 kV/mm for Teflon. How can the authors ensure that there is no dielectric breakdown during the experiment? How to make sure that the measured results are not from capacitive forces?

If my initially suggested experiments are not possible, an alternative could be the frequency dependent measurement, since materials' piezoelectric properties are not strongly dependent on the V_{ac} frequency, but other effects could be frequency dependent. Using different probes, including stiffer ones, could also help validate the measured piezoelectric constants. As suggested, the authors should, at least, measure the peptide crystal's piezoelectric response across a range of V_{ac} frequency (from ~ 1 kHz to ~ 100 kHz) which is doable for most PFM setups.

The authors mention that "The measured samples are single crystals and do not require any additional polarization as they crystallize in non-centrosymmetric space groups and so are inherently piezoelectric without requiring polling." I noticed that peptide crystals are randomly placed on substrates (Fig.S11 and S12), and thus the polarization should not be always aligned in the direction of the applied electric field (Fig.S5). The authors should explain how to know crystals' orientation, why they don't see any negative value of "piezoreponse current", and how to correlate the measured results to those from simulations.

Author Response to the reviewer comments for “Molecular Engineering of Piezoelectricity in Collagen-Mimicking Peptide Assemblies”

Reviewer 1

The revised manuscript addressed well for the critiques that this reviewer raised. Therefore, current manuscript can be publishable for Nature Comm.

Authors' response:

We thank the reviewer for the recommendation of acceptance our manuscript for publication.

Reviewer 2

1. Reviewer's comment:

I appreciate the authors' efforts to address my concerns and comments. I totally understand the difficulty in probing nanoscale materials' piezoelectric properties. However, since the piezoelectricity of peptide crystals is a major claim of this paper, I do think that it is necessary to clarify the origin of these measured piezoelectric properties, especially those measured by PFM. Thus, I cannot recommend publication before addressing the following issues.

Authors' response:

We very much appreciate the referee's continued support, understanding, and interest in our work. We are happy to further clarify the origin of the piezoelectric properties that was measured by us using PFM. As described below, we have completely addressed all the issues raised by the referee. We are very grateful to the referee for proposing the additional PFM experiments and data interpretations to verify and quantify the piezoelectric effects, which we now include in our revised manuscript.

Reviewer's comment:

To be clear, my previous comments suggest additional PFM experiments and data interpretations to verify and quantify the piezoelectric effects, rather than ask the authors to proof materials' ferroelectric nature. As mentioned by the authors, the suggested articles, including Gruverman, Nature Communications 10, 1661 (2019), do give reasons why current PFM methods are not reliable for piezoelectricity measurements, including the voltage-induced deformation. Thus, using a probe with a certain stiffness at a fixed frequency will not eliminate

other effects. For the same reason, together with the difference in surface charge, sizes, and shapes, the positive and negative controls cannot ensure the accuracy of the measurements.

I understand the difficulty in measuring nanoscale crystals that are movable on substrates, but the PFM experiment, together with the very low g_{33} from the nanogenerator, raise a lot of concerns for me. For example, the authors applied up to 20 V across a ~ 180 nm (Fig.1d) thick peptide crystal in the ambient environment. Based on my quick calculation, the resulting electric field reaches 111 kV/mm, which is extremely high comparing to breakdown electric fields of ~ 3 kV/mm for dry air and ~ 60 kV/mm for Teflon. How can the authors ensure that there is no dielectric breakdown during the experiment? How to make sure that the measured results are not from capacitive forces?

If my initially suggested experiments are not possible, an alternative could be the frequency dependent measurement, since materials' piezoelectric properties are not strongly dependent on the V_{ac} frequency, but other effects could be frequency dependent. Using different probes, including stiffer ones, could also help validate the measured piezoelectric constants. As suggested, the authors should, at least, measure the peptide crystal's piezoelectric response across a range of V_{ac} frequency (from ~ 1 kHz to ~ 100 kHz) which is doable for most PFM setups.

Authors' response:

We thank the reviewer for motivating us to further question and substantiate the origin of the electromechanical response as measured by PFM, as we agree it can be difficult to determine the true origin of the signal. We have performed the suggested checks for frequency dependence in our measurements, by measuring the piezoelectric response of the peptide crystals across a range of V_{ac} frequency (from ~ 1 kHz to ~ 100 kHz). We now include in the updated *Supporting Information* this additional dataset showing the frequency dependence of the piezoresponse signal in the range from 1 kHz to 100 kHz. These plots clearly show that the signal is not frequency dependent around the chosen low frequency of 20 kHz. The large fluctuations at very low frequencies (< 2 kHz) are due to the inherent system limitations of the feedback loop and lock-in amplifier. All frequency measurements were acquired using a stiff doped-diamond cantilever with a spring constant of 40 N/m with 20 V applied voltage. The magnitude of the stable frequency independent piezoresponse signal is close to that initially measured in all crystals, verifying that the magnitude of the piezoresponses reported in this

manuscript is both independent of the applied frequency and independent of the probe stiffness. These control experiments confirm that our reported measurements quantify a genuine piezoelectric response, free from unwanted parasitic or additive contributions.

Changes made to the manuscript:

Page 13, line 14-17:

“The linear relationship between the piezoresponse as measured by the photodiode system and applied voltage, and also the minimal frequency dependence of output piezoresponse is indicative of a genuine piezoelectric property.”

Supporting Information page 18, line 4-12:

“Supplementary Fig. 14-16 show the measured frequency dependence of the piezoelectric responses for both the Pro-Phe-Phe and Hyp-Phe-Phe tripeptide crystals. Supplementary Fig. 14 depicts the frequency dependence of the vertical piezoresponse for Pro-Phe-Phe crystals while Supplementary Figs. 15 and 16 respectively show the frequency dependence of the vertical and lateral signals for Hyp-Phe-Phe. All frequency spectra were acquired using a stiff, diamond coated probe with a spring constant of 40 N/m at 20 V applied voltage. Minimal frequency dependence is observed with no significant variations in the smooth piezoresponse present in the region of 20 kHz. This is demonstrative of a genuine piezoelectric response, converged with respect to the frequency of the applied voltage.”

Supplementary Fig. 14: Measured frequency dependence of the vertical piezoresponse for Pro-Phe-Phe.

Supplementary Fig. 15: Measured frequency dependence of the vertical piezoresponse for Hyp-Phe-Phe.

Supplementary Fig. 16: Measured frequency dependence of the lateral piezoresponse for Hyp-Phe-Phe.

2. Reviewer's comment:

The authors mention that “The measured samples are single crystals and do not require any additional polarization as they crystallize in non-centrosymmetric space groups and so are inherently piezoelectric without requiring polling.” I noticed that peptide crystals are randomly placed on substrates (Fig.S11 and S12), and thus the polarization should not be always aligned in the direction of the applied electric field (Fig.S5). The authors should explain how to know crystals' orientation, why they don't see any negative value of “piezoresponse current”, and how to correlate the measured results to those from simulations.

Authors' response:

Regarding the orientation of the crystals, we are happy to clarify that we measure multiple sites on multiple crystals in random orientations, which allows to obtain a distribution of the average piezoelectric response. This is done to gauge the size of response that could be obtained from an assembly of these crystals in a device such as a power generator, the construction of which is greatly simplified when we do not require perfectly aligned crystals to generate a technologically useful level of response. Due to the large number of non-zero piezoelectric constants possessed by these crystals (please see Fig. 2a, b), applying an electric field in any direction along the crystals will stimulate a piezoelectric deformation, which is detected by the

PFM probe. Due to the fact that PFM applies an AC field to the crystals under test, we will never measure negative values for the piezoresponse. The effective piezoresponse magnitude is always reported. To be clear, the piezoresponse current we report in nA is the unit of measurement used by the AFM photodiode, which we then convert to a deformation to obtain the piezoelectric constants (please see Section 3 of SI). This is not to be confused with a current generated by the crystal due to its piezoelectricity, which we do not attempt to measure. We do not claim that we are probing an exact individual piezoelectric coefficient as predicted by modelling. Instead, we compare the average response to the relevant piezoelectric tensor components (please see Table 1, main text).

REVIEWERS' COMMENTS

Reviewer #2 (Remarks to the Author):

The revision satisfies my previous comments. I would suggest publication as is.

Author Response to the reviewer comments for “Molecular Engineering of Piezoelectricity in Collagen-Mimicking Peptide Assemblies”

Reviewer 2

The revision satisfies my previous comments. I would suggest publication as is.

Authors' response:

We thank the reviewer for the recommendation of publication of our manuscript.